# Isometric Regularization for Manifolds of Functional Data

**Hyeongjun Heo**[1]**, Seonghun Oh**[2]**, Jae Yong Lee**[3]**, Young Min Kim**[1*]**, Yonghyeon Lee**[4*]
[1]Seoul National University    [2]Yonsei University    [3]Chung-Ang University
[4]Korea Institute for Advanced Study
{heo0224, youngmin.kim}@snu.ac.kr
rendell@yonsei.ac.kr   jaeyong@cau.ac.kr   ylee@kias.re.kr

## Abstract

While conventional data are represented as discrete vectors, Implicit Neural Representations (INRs) utilize neural networks to represent data points as continuous functions. By incorporating a shared network that maps latent vectors to individual functions, one can model the distribution of functional data, which has proven effective in many applications, such as learning 3D shapes, surface reflectance, and operators. However, the infinite-dimensional nature of these representations makes them prone to overfitting, necessitating sufficient regularization. Naïve regularization methods – those commonly used with discrete vector representations – may enforce smoothness to increase robustness but result in a loss of data fidelity due to improper handling of function coordinates. To overcome these challenges, we start by interpreting the mapping from latent variables to INRs as a parametrization of a Riemannian manifold. We then recognize that preserving geometric quantities – such as distances and angles – between the latent space and the data manifold is crucial. As a result, we obtain a manifold with minimal intrinsic curvature, leading to robust representations while maintaining high-quality data fitting. Our experiments on various data modalities demonstrate that our method effectively discovers a well-structured latent space, leading to robust data representations even for challenging datasets, such as those that are small or noisy.

## 1 Introduction

Implicit Neural Representations (INRs), often referred to as Neural Fields, are functional representations of data points typically expressed as $f_\theta : \mathcal{X} \to V$ with network parameters $\theta$, where $\mathcal{X}$ is an input space such as spatial or temporal coordinates and $V$ is an output vector space (Xie et al., 2022). One significant advantage of this representation is that, since individual data points are continuous functions, one can sample values of $V$ at an arbitrary resolution from $\mathcal{X}$. Moreover, given multiple instances of functions, it has recently been shown that, instead of learning separate network weights for each data instance, a shared neural network can be constructed by conditioning on an auxiliary latent variable $z$:

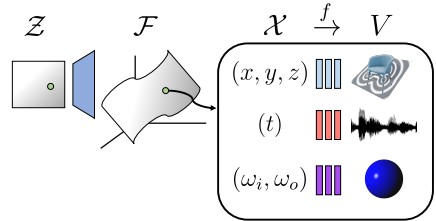

Figure 1: Illustrations of manifolds of functional data.

$$F : \mathcal{X} \times \mathcal{Z} \to V, \tag{1}$$

where, for an instance of a latent vector $z \in \mathcal{Z}$, $F_z := F(\cdot, z)$ corresponds to a data point of functional representation. This latent variable model has been widely applied to tasks such as data compression, generation, prediction, and solving differential equations, across various data types, including images (Sitzmann et al., 2020; Tancik et al., 2020), 3D shapes (Park et al., 2019), textures (Fan et al., 2022), and functions for differential equations (Lu et al., 2019; 2021).

---

*Corresponding authors.
Project website: heo0224.github.io/IRMF_projectpage

To achieve strong performance in downstream tasks such as interpolation and reconstruction, the latent variable model $F$ must satisfy two essential criteria: (i) it should accurately fit the data instances, resulting in high-fidelity reconstructions, and (ii) the latent space $\mathcal{Z}$ should be well-behaved, ensuring that small changes in the latent space lead to gradual and predictable changes in the output. Without proper regularization, particularly when the training dataset is small or noisy, simply fitting the data instances can cause the latent space to become ill-behaved, leading to poor representations and weaker performance in downstream tasks. Recently, Lipschitz regularization (Liu et al., 2022) proposes to learn a well-behaved latent space with a continuous function by promoting smoothness in $F$. However, they regularize across both input domains, $\mathcal{X}$ and $\mathcal{Z}$, and may result in over-smoothing of data instances in $\mathcal{X}$, failing to meet the aforementioned two criteria simultaneously.

In this paper, we propose a balanced regularization method that produces a well-behaved latent space $\mathcal{Z}$ without overly smoothing the data in $\mathcal{X}$. Our approach begins with a geometric interpretation of the latent variable model $F$ as a Riemannian manifold of functional data, where the latent variable $z$ corresponds to the coordinates of the manifold, as depicted in Figure 1. Based on this interpretation, we regularize $F$ in the $z$ component to produce a smooth and parsimonious manifold – resulting in small intrinsic curvatures (Lee & Park, 2023) and well-behaved latent spaces – while leaving the $x$ component unregularized, thus preserving the fidelity of individual data instances.

Specifically, we adopt the recent isometric regularization method developed for finite-dimensional data (Lee et al., 2022b), which preserves geometric quantities such as distances and angles between the latent space and the manifold, and extend it to infinite-dimensional functional data. While regularizing the manifold of discretized data on a grid, like images, has been widely studied (Chen et al., 2020; Lee et al., 2021; 2022b; Jang et al., 2023; Lee et al., 2022a; Nazari et al., 2023; Lee & Park, 2023), applying these methods to latent variable INRs is challenging due to variability in sample locations and numbers, making consistent regularization difficult across different data instances. To address this, we propose a discretization-agnostic approach to isometric regularization for $F$. One notable difference from the finite-dimensional cases is that this approach involves integrating with respect to a positive measure in $\mathcal{X}$, which requires significant computation. To make this practical, we develop an approximate yet efficient algorithm, using methods such as Hutchinson's trace estimator and Monte Carlo approximation.

Through extensive experiments, we validate the effectiveness of our isometric regularization for functional data on various tasks with INRs including neural Signed Distance Functions (SDFs) (Park et al., 2019), neural Bidirectional Reflectance Distribution Functions (BRDFs) (Fan et al., 2022), and Deep Operator Networks (DONets) (Lu et al., 2019) showing that our method is modality-independent. Further, we illustrate that isometric regularization guides the model $F$ to learn an accurate manifold with smooth latent space leading to good generalization performance and robustness to noise in data.

## 2 RELATED WORK

**Latent Variable Models for Functional Data.** Recently, many works have shown that continuous signals can be efficiently modeled as a function parameterized by a neural network, referred to as Implicit Neural Representations (INRs). INRs directly map the input variable (i.e., spatial coordinate or time index) into the corresponding value and efficiently represent a broad class of high-resolution data including images (Ha, 2016), audios (Sitzmann et al., 2020), videos (Chen et al., 2022; Li et al., 2021), 3D shapes with SDF (Park et al., 2019), occupancy (Mescheder et al., 2019), and radiance fields (Mildenhall et al., 2020). INRs are also actively employed to represent solutions to diverse differential equations in physics-informed machine learning (Raissi et al., 2019; Karniadakis et al., 2021; Lu et al., 2019; 2021). Instead of representing a single function, recent INR works have shown learning the distribution of a range of functional data by conditioning a shared neural network on a latent variable $z$ for each functional data (Chen & Zhang, 2019; Park et al., 2019; Mescheder et al., 2019; He et al., 2022; Fan et al., 2022; Du et al., 2021). There are two approaches to constructing a latent variable model. One straightforward way to condition $z$ is to concatenate input coordinate $x$ with $z$ (Park et al., 2019; Fan et al., 2022; He et al., 2022). Alternatively, novel network architectures are used to condition the latent variable such as hypernetworks (Du et al., 2021; Gokbudak et al., 2024; Lee et al., 2023) or attention networks (Rebain et al., 2022). We emphasize that our framework is generic and can be applied to both architectures.

**Geometric Regularization for Manifold Learning.** For finite-dimensional data, one can interpret the learned latent space of generative models as an explicit parametrization of the data manifold (Arvanitidis et al., 2017; Lee, 2023). It has been widely demonstrated that proper regularization on latent space can significantly improve the performance on downstream tasks for generative models, such as clustering, interpolation, retrieval, and more (Chen et al., 2020; Lee et al., 2021; 2022b; Jang et al., 2023; Lee et al., 2022a; Nazari et al., 2023; Lee & Park, 2023; Lim et al., 2024; Lee, 2024).

However, it is less explored to extend such manifold interpretation of latent variables beyond finite-dimensional data. While Du et al. (2021) view the latent variable model as a manifold of INRs and presents a local isometry loss, they require functions to be sampled from fixed points in coordinates with a fixed resolution. However, functional data often needs different sampling strategies for each data instance as it considerably affects the fitting quality (Park et al., 2019; Sztrajman et al., 2021). Furthermore, it is impossible to sample from fixed points with fixed numbers for real data in certain scenarios, for example, mobile sensors such as floating buoys change their location over time and some sensors can malfunction (Luo et al., 2024). Instead, we design geometric regularization method specifically for functional data without fixing sampling condition.

## 3 Riemannian Manifold of Functional Data

In this section, we introduce a geometric view that considers latent variable INRs as a Riemannian manifold of functional data, laying the foundation for subsequent isometric regularization. We start with a formal definition of the space of functional data $f : \mathcal{X} \to V$, where $\mathcal{X} = \mathbb{R}^n$ and $V$ is a vector space with the standard inner product, such that the inner product between $v, w \in V$ is $v^T w$. Throughout, we will consider a set of square-integrable mappings

$$\mathcal{F} := \{f : \mathcal{X} \to V \mid \int f(x)^T f(x) \, dx < \infty\} \tag{2}$$

as a *functional data space*, which is an infinite dimensional vector space[1].

### 3.1 Inner Products on Functional Data Space

First, we define the inner product on the functional data space $\mathcal{F}$. Let $\delta_1, \delta_2$ be square-integrable functions with a countably additive measure $\mu$ (e.g., probability measure) in $\mathcal{X}$. A standard way to define an inner product between them is as follows:

$$\langle \delta_1, \delta_2 \rangle := \int \delta_1(x)^T \delta_2(x) \, d\mu(x). \tag{3}$$

This inner product does not depend on a functional data point $f \in \mathcal{F}$. However, in some cases, we need to consider different measures for each functional data point, as important regions in $\mathcal{X}$ could be different depending on $f$. For example, DeepSDF (Park et al., 2019) utilizes truncated SDF (TSDF) that clamps distance values of regions far from the surface to be constant, which means that function values only from areas close to the surface have meaningful information. In such a case, we would like to use a measure $\mu$ concentrated near the surface depending on $f$. Therefore, in this work, we consider the inner product to depend on the function $f$ by employing a function-dependent measure $\mu_f$,

$$\langle \delta_1, \delta_2 \rangle_f := \int \delta_1(x)^T \delta_2(x) \, d\mu_f(x) \quad \text{for} \quad f \in \mathcal{F}. \tag{4}$$

### 3.2 Riemannian Manifold Embedded in Functional Data Space

Given our definition of the functional data space $\mathcal{F}$ in Equation (2), we can interpret the model $F : \mathcal{X} \times \mathcal{Z} \to V$ as a mapping from the latent space $\mathcal{Z}$ to the functional data space $\mathcal{F}$:

$$\mathrm{h} : \mathcal{Z} \to \mathcal{F} \quad \text{s.t.} \quad z \mapsto \mathrm{h}(z) := F_z, \tag{5}$$

where $F_z$ is a functional data and $F_z(x) := F(x, z)$. Taking DeepSDF as an example, a latent code $z$ is mapped to a 3D shape represented by SDF $F_z : \mathbb{R}^3 \to \mathbb{R}$, which takes a spatial point

---

[1] $\mathcal{F}$ is a Banach space, a complete normed vector space, with respect to the Lebesgue measure.

$x \in \mathcal{X} = \mathbb{R}^3$ as input and outputs the signed distance value $d \in V$. Moving $z$ in the latent space leads to a continuous change in the output signed distance of $F_z(x)$.

If $\mathcal{Z}$ is an $m$-dimensional manifold (e.g., $\mathbb{R}^m$), then the image of h, denoted by $h(\mathcal{Z}) = F(\cdot, \mathcal{Z})$, is an $m$-dimensional manifold embedded in the functional data space $\mathcal{F}$ under some mild conditions[2], as illustrated in Figure 1. We note that, although $\mathcal{F}$ is infinite-dimensional, the embedded manifold is finite-dimensional. This manifold interpretation offers a geometric understanding of the model $F$, allowing us to leverage tools from differential geometry to develop regularization in a principled manner, as we demonstrate in the subsequent section.

We conclude this section by proposing a Riemannian geometric structure for the embedded manifold. Given the function-dependent inner product $\langle \cdot, \cdot \rangle_f$ for the functional data space $\mathcal{F}$ in Equation (3), a standard way to define a Riemannian metric for the embedded manifold is via projecting $\langle \cdot, \cdot \rangle_f$ to the manifold. If $\mathcal{Z} = \mathbb{R}^m$ is treated as a local coordinate space, then the projected Riemannian metric can be expressed in $\mathcal{Z}$ as follows:

$$ds^2 = \sum_{i,j=1}^{m} \int \left( \frac{\partial F(x,z)}{\partial z^i} \right)^T \frac{\partial F(x,z)}{\partial z^j} d\mu_{F_z}(x) dz^i dz^j, \tag{6}$$

where the integral $h_{ij}(z) := \int \left( \frac{\partial F(x,z)}{\partial z^i} \right)^T \frac{\partial F(x,z)}{\partial z^j} d\mu_{F_z}(x)$ forms an $m \times m$ positive-definite matrix $H(z) = (h_{ij})_{i,j=1}^m$ called a Riemannian metric. We note that, with this metric, given an infinitesimal change in the latent value $z \mapsto z + dz$, the length of $dz$ is computed via $dz^T H(z) dz = \int (dF(x,z))^T dF(x,z) d\mu_{F_z}(x)$, where $dF(x,z) \approx F(x, z+dz) - F(x,z)$.

## 4 ISOMETRIC REGULARIZATION

Without proper regularization, the latent space can become ill-behaved, overfitting to the data instances, which is exacerbated by the infinite dimensionality of the function space. Our goal of finding a well-behaved latent space without compromising data fidelity can be restated in geometric terms: the given data instances lie in the manifold $h(\mathcal{Z})$ while the mapping h from $\mathcal{Z}$ to the manifold $h(\mathcal{Z})$ should preserve geometry as much as possible. The former condition of fitting data instances is enforced by a task-relevant loss function, which varies depending on the application. Examples in the experiment section include the signed distance and the reflectance fitting loss in deep generative models and the regression loss in operator learning. The latter geometry-preserving condition can be achieved through our proposed isometric regularization.

By 'geometry-preserving,' we mean that distances and angles measured in the manifold should be preserved in the latent space. Consequently, slight changes in latent values will not result in dramatic changes in the output data but will cause changes of a consistent magnitude, leading to a well-behaved latent space. Formally, let the latent space $\mathcal{Z} = \mathbb{R}^m$ be assigned with the identity metric and the manifold of functional data be assigned with the projected Riemannian metric in Equation (6). Throughout, we will denote the Jacobian of $F$ by $J(x,z) = \frac{\partial F(x,z)}{\partial z} \in \mathbb{R}^{\dim(V) \times m}$ and the projected metric

$$H(z) = \int J(x,z)^T J(x,z) d\mu_{F_z}(x) \in \mathbb{R}^{m \times m}. \tag{7}$$

A mapping h is called an isometry if $I = H(z)$ for all $z \in \mathcal{Z}$. If $F$ satisfies the above condition, then for infinitesimal changes in the latent value $dz$ and the corresponding change in the data $dF(x,z) \approx F(x, z+dz) - F(x,z)$, the norm of $dz$ in the latent space is equal to the norm of $dF$, i.e., $dz^T dz = dz^T H(z) dz = \int dF^T dF d\mu_{F_z}(x)$, meaning that $F$ preserves local distances and angles. If this holds for all $z$, then $F$ preserves the global geometry. As discussed in Lee et al. (2022b), we encourage a scaled isometry of $F$ that preserves angles and scaled distances, such that $I = cH(z)$ for some positive scalar $c$ and all $z$.

It is important to note that enforcing this scaled isometry condition does not impose a smoothness requirement on the $x$-coordinates, unlike Lipschitz regularization. For instance, consider the limiting

---

[2]The mapping h $: \mathcal{Z} \to \mathcal{F}$ is an embedding if it is injective immersion. In other words, (i) (injectivity) if $F(x, z_1) = F(x, z_2)$, then $z_1 = z_2$ for all $x \in \mathcal{X}$ and (ii) (immersion) if $\frac{\partial F(x,z)}{\partial z} v = 0$ for $v \in \mathbb{R}^m$, then $v = 0$ for all $x, z$.

case where (i) $I = cH(z)$ is exactly satisfied for all $z$ and (ii) $d\mu_{F_z} = d\mu$ is independent of $z$. In this scenario, a trivial solution is that $J(x, z)$ remains constant over $z$. Crucially, no conditions are imposed on the $x$-coordinates. This is one of the key properties of our regularization, which results in a well-behaved latent space without compromising data fidelity.

**Relaxed Distortion Measure.** We introduce a functional called the relaxed distortion measure to impose *isometric regularization* for manifolds of functional data. It quantifies how far a mapping $F$ is from being a scaled isometry. To handle the continuous data space, we apply an approximation based on efficient sampling algorithm. Here, we provide formulas for the relaxed distortion measure with minimal details; please refer to Appendix A for more information and proofs. Throughout, we will use a probability measure for $\mu_{F_z}$ in Equation (7) and denote its density function by $p(x; F_z)$. Then the Riemannian metric can be written as $H(z) = \mathbb{E}_{x \sim p(x;F_z)}[J^T(x, z)J(x, z)] \in \mathbb{R}^{m \times m}$. Let $\lambda_i(z)$ be the eigenvalues of $H(z)$ for $i = 1, \ldots, m$.

An equivalent characterization of the scaled isometry condition exists: if $\lambda_i(z) = c$ for some positive scalar $c$ and all $i$ and $z$, then $F$ is a scaled isometry. Enforcing this condition for all $z \in \mathbb{R}^m$ is unnecessary since latent points only occupy a specific region. We introduce a latent probability density $P_{\mathcal{Z}}$ and focus on its support. Then, a coordinate-invariant relaxed distortion measure $\mathcal{G}(F, P_{\mathcal{Z}})$ for $F$ with respect to $P_{\mathcal{Z}}$ can be defined as follows:

$$\mathcal{G}(F, P_{\mathcal{Z}}) := \mathbb{E}_{z \sim P_{\mathcal{Z}}}\Big[\sum_{i=1}^{m} \Big(\frac{\lambda_i(z)}{\mathbb{E}_{z \sim P_{\mathcal{Z}}}[\sum_{i=1}^{m} \lambda_i(z)/m]} - 1\Big)^2\Big], \tag{8}$$

which is minimal if and only if $F$ meets the scaled isometric condition in support of $P_{\mathcal{Z}}$. Ignoring a constant additive term, $\mathcal{G}(F, P_{\mathcal{Z}})$ becomes a scalar multiple of

$$\frac{\mathbb{E}_{z \sim P_{\mathcal{Z}}}[\text{Tr}(\big(\mathbb{E}_{x \sim p(x;F_z)}[J^T(x, z)J(x, z)]\big)^2)]}{\mathbb{E}_{z \sim P_{\mathcal{Z}}}[\text{Tr}(\mathbb{E}_{x \sim p(x;F_z)}[J^T(x, z)J(x, z)])]^2}. \tag{9}$$

When training manifolds of functional data $F$, we add the term above to the original loss term to incorporate *isometric regularization* for the latent mapping. The expression in Equation (9) is similar to the one introduced in Lee et al. (2022b), but ours involves $\mathbb{E}_{x \sim p(x;F_z)}$ in the traces, and the Jacobian also depends on $x$. This difference makes applying existing implementations non-trivial, motivating us to develop a new approximate and efficient distortion computation algorithm.

**Efficient Approximation of Eq. (9).** The computation of Equation (9) involves (i) calculating the Jacobian $J(x, z)$ and (ii) computing the expectation of $J^T J$ with respect to $p(x; F_z)$. These calculations must be performed at each training iteration, which significantly slows down the overall process. To address this, we introduce several practical techniques for approximate computation. First, we estimate the trace terms using Hutchinson's stochastic trace estimator (Hutchinson, 1989), $\text{Tr}(A) = \mathbb{E}_{v \sim \mathcal{N}(0,I)}[v^T A v]$, which allows us to bypass the need to compute the full Jacobian $J(x, z)$ and instead focus on Jacobian-vector and vector-Jacobian products. Second, to compute the expectation of $J^T J$ with respect to $p(x; F_z)$, we typically need to sample from $p(x; F_z)$. However, the sampling distribution $p(x; F_z)$ evolves during the training of $F$, making the online sampling infeasible. To mitigate this, we use offline samples from $p(x; F_i^*)$, where $F_i^*$ represents the functional data for training $z_i$.

---

**Algorithm 1:** Efficient approximation of Eq. (9)

**Precondition:** input concatenation $(F : \mathbb{R}^{n+m} \to \mathbb{R}^l)$
**Input:** latent codes $\{\mathbf{z}_0, \ldots, \mathbf{z}_N\}$ & input coordinate samples
$\{\{\mathbf{x}_0^{(0)}, \ldots, \mathbf{x}_0^{(K)}\}, \ldots, \{\mathbf{x}_N^{(0)}, \ldots, \mathbf{x}_N^{(K)}\}\}$
**Output:** Relaxed distortion measure $\mathcal{G}$

1   $\mathcal{G}_1, \mathcal{G}_2 \leftarrow 0$
2   Augment $\mathbf{z}$ with the modified mix-up data-augmentation
3   **forall** $\mathbf{z}_i$ *in* $\mathbf{z}$ **do**
4     $\mathbf{x}_i \leftarrow \{\mathbf{x}_i^{(0)}, \ldots, \mathbf{x}_i^{(K)}\}$
5     Sample vector $v_i \sim \mathcal{N}(0, I_{m \times m})$
6     Expand $v_i$ by repeating $K$ times
7     Augment vector $v_i$ by concatenating $[\vec{0}_{k \times n}, v_i]$
8     Compute $\text{G} = J(\mathbf{x}_i, \mathbf{z}_i)v_i$ with Jacobian-vector product
9     $\mathcal{G}_1 \leftarrow \mathcal{G}_1 + \mathbb{E}_z[\mathbb{E}_x[\text{G}^T\text{G}]]$
10    Compute $\text{D} = \text{G}^T \partial F(\mathbf{x}_i, \mathbf{z}_i)/\partial(x, z)$ with vector-Jacobian product
11    Slice the index of $\text{D}$ by taking the last $m$-th components
12    $\mathcal{G}_2 \leftarrow \mathcal{G}_2 + \mathbb{E}_z[\mathbb{E}_x[\text{D}]^T\mathbb{E}_x[\text{D}]]$
13   **end**
14   $\mathcal{G} \leftarrow \mathcal{G}_2/\mathcal{G}_1$
15   **return** $\mathcal{G}$

---

We also apply the latent augmentation method from Chen et al. (2020); Lee et al. (2022b) to define $P_{\mathcal{Z}}$. A pseudocode for the approximate computation of the relaxed distortion measure is provided in Algorithm 1; more details can be found in Appendix B.

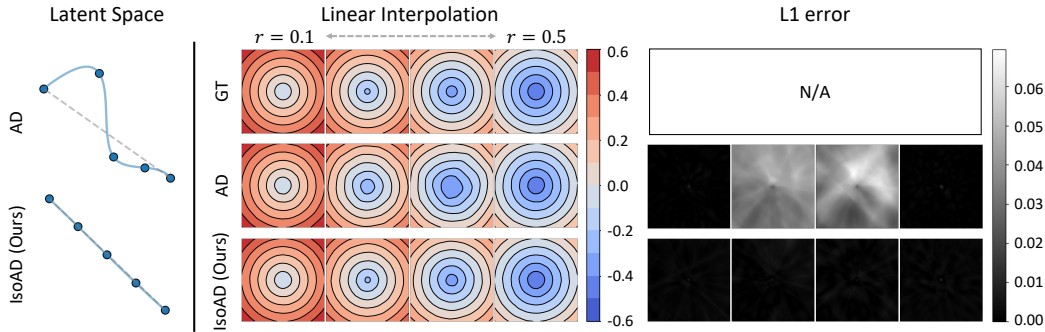

Figure 2: Toy example of neural SDFs with five circles. (Left) Two-dimensional latent space of auto-decoder trained with and without isometric regularization. (Right) SDFs decoded from linear interpolants & L1 errors on SDFs.

## 5 EXPERIMENTS

In this section, we conduct extensive experiments to show the effect of isometric regularization on three data modalities: neural SDFs, neural BRDFs, and neural operators.

### 5.1 NEURAL SDFS

SDF is a function $f : \mathbb{R}^n \to \mathbb{R}$ that maps a spatial point to its signed distance to the surface of the shape. The surface points of the shape are extracted from the zero-level set of $f$. We train the neural SDFs with the latent variable model, denoted by $F_\theta : \mathcal{X} \times \mathcal{Z} \to \mathbb{R}$, where $x \in \mathcal{X}$ represents spatial coordinates, $z \in \mathcal{Z}$ is a shape latent vector, and $F_\theta(\cdot, z)$ is a SDF value, following the auto-decoder in DeepSDF (Park et al., 2019). We provide experimental details in Appendix C.1.

#### 5.1.1 TOY EXAMPLES ON SIMILAR SHAPES

We first demonstrate the effect of isometric regularization with simple 2D shapes. We train auto-decoders using a two-dimensional latent space $z \in \mathbb{R}^2$ without any regularization (AD) and with isometric regularization (IsoAD), and compare the results. Appendix D.1 contains additional analysis on the probability density $p(x; F_z)$.

**Dataset.** We make training datasets with three kinds of simple shapes: circles, squares, and equilateral triangles. Each dataset contains $N$ similar shapes whose parameters change linearly, for example, circles with radius $r$ from 0.1 to 0.5 or squares with side lengths from 0.1 to 0.5. Then, we train the network with the SDF values of each shape sampled from a $32 \times 32$ grid.

Table 1: L1 errors ($\times 10^{-3}$) on linear interpolation.

|  | $N$ | AD | IsoAD (Ours) |
|---|---|---|---|
| Circle | 5 | 36.5885 | **2.7497** |
|  | 10 | 10.1056 | **2.5622** |
|  | 20 | 9.5139 | **1.9630** |
| Square | 5 | 13.1302 | **6.7625** |
|  | 10 | 2.2039 | **1.9842** |
|  | 20 | 2.7227 | **1.4770** |
| Triangle | 5 | 21.8526 | **4.4000** |
|  | 10 | 2.0579 | **1.7629** |
|  | 20 | 2.7957 | **1.3722** |

**Results.** Given the smallest and the biggest shapes for each model, we generate intermediate shapes by linearly interpolating the latent space. We evaluate the decoded intermediate shapes against reference shapes created by interpolating the shape parameters. We compute the L1 errors at $256 \times 256$ grid points. Quantitative results in Table 1 show that isometric regularization significantly improves the interpolation results, especially when the number of training data $N$ is small. Figure 2 shows the latent space of auto-decoders trained with five circles. When we linearly increase the circle's radius, the most intuitive latent manifold would be a straight line with equidistant latent codes. While auto-decoder without regularization leads to a distorted latent space resulting in undesirable interpolation results even with extremely simple shapes, isometric regularization encourages the output shape to expand with constant velocity leading to predictable interpolation results.

Table 2: Quantitative results on surface reconstruction.

| Dataset | MNIST | | | | ShapeNet | | | |
|---|---|---|---|---|---|---|---|---|
| $N$ | 100 | | 500 | | 271 | | 542 | |
| Metrics | L1 error average | L1 error median | L1 error average | L1 error median | CD average | CD median | CD average | CD median |
| DeepSDF | 0.0366 | 0.0355 | 0.0288 | 0.0285 | 0.002202 | 0.001812 | 0.001588 | 0.001207 |
| LipDeepSDF | 0.0351 | 0.0341 | 0.0279 | 0.0255 | 0.001999 | 0.001606 | 0.001361 | 0.001054 |
| IsoDeepSDF (ours) | **0.0283** | **0.0265** | **0.0267** | **0.0245** | **0.001915** | **0.001472** | **0.001263** | **0.000925** |

### 5.1.2 SURFACE RECONSTRUCTION

We now train neural SDFs with a variety of more complex 2D & 3D shapes from MNIST (Deng, 2012) and ShapeNet (Chang et al., 2015). Similar to Section 5.1.1, we use auto-decoder architecture for training 2D SDFs. For training the autoencoder with 3D SDFs, we use an encoder from Pointnet Qi et al. (2017) and a decoder from DeepSDF, following the experimental setup from Liu et al. (2022).

We can then reconstruct the surfaces by finding latent codes that minimize the output SDF values at the surface points provided at the test time. Test time optimization of a latent code is commonly used in auto-decoders and autoencoders to reconstruct a surface from the point cloud data (Park et al., 2019; Gurumurthy & Agrawal, 2019; Liu et al., 2022). In particular, latent code optimization is needed when a simple pass forward through the autoencoder leads to unsatisfying results. We compare our method, referred to as IsoDeepSDF, with vanilla DeepSDF without regularization and DeepSDF with Lipschitz regularization (LipDeepSDF) (Liu et al., 2022).

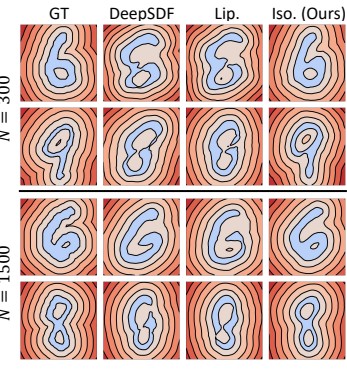

Figure 3: Qualitative results of surface reconstruction given zero-level set on MNIST.

**MNIST Dataset.** We make two datasets with different numbers of training data $N = 300, 1500$. The datasets contain 100 & 500 images randomly chosen from each of three digits [6, 8, 9]. The images are transformed to 2D signed distance fields on a $64 \times 64$ grid, where the contour of the digit is the zero-level set. For test time optimization, we randomly sample 256 points from zero-level surfaces of the test dataset with 100 images from each digit [6, 8, 9].

**ShapeNet Dataset.** We randomly choose 5% ($N = 271$) and 10% ($N = 542$) of shapes from the chair category of ShapeNetV2 for training datasets. As we observe that the choice of training dataset strongly impacts the reconstruction results, we make five different datasets for each $N$ random choice. Quantitative results are the average of the metrics evaluated from each dataset. The test-time optimization reconstructs the full 3D shape from partial point clouds obtained by deleting the right half of the surface point cloud.

**Results.** We compute the L1 error for 2D SDFs on a $256 \times 256$ grid and Chamfer distance (CD) for 3D shapes with 30,000 points on the surfaces for evaluation metrics. Quantitative results are summarized in Table 2. IsoDeepSDF quantitatively outperforms the others for all datasets, demonstrating that isometric regularization helps to learn a well-behaved latent space, leading to better reconstruction results. While decreasing $N$ severely deteriorates the reconstruction quality on the MNIST dataset, isometric regularization significantly reduces the performance degradation, providing reliable regularization despite scarce data points. Figure 3 and Figure 4 each show the qualitative results of surface reconstruction on the MNIST and ShapeNet datasets. Our method qualitatively shows better reconstruction results for both cases with 2D point clouds and 3D partial point clouds. Especially for 3D shapes, DeepSDF fails to fully reconstruct the unseen parts of the chairs, while our method can better reconstruct the overall shape given partial observations.

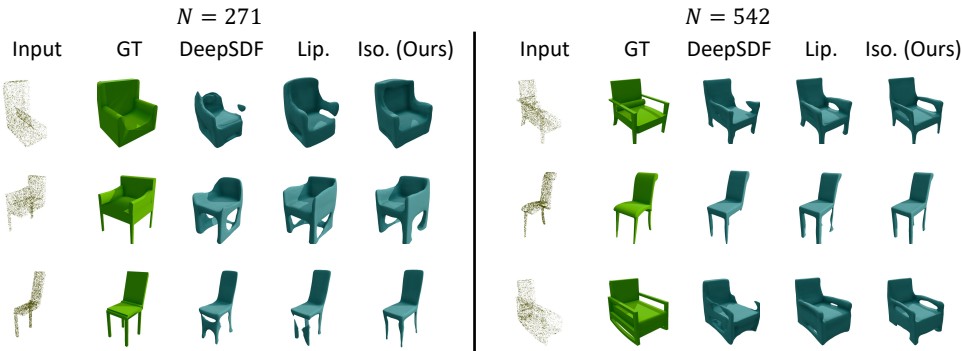

Figure 4: Qualitative results of surface reconstruction on ShapeNet chair dataset. Our method helps to learn better latent space, leading to better reconstruction results from partial observation.

## 5.2 NEURAL BRDFS

The bidirectional reflectance distribution function (BRDF) is a function that takes the incoming and outgoing directions of light as input and outputs the reflected radiance on the surface, which is used to render the appearance of materials. Recent works have proposed generalizable BRDF representations with latent variable models to reconstruct a BRDF of unseen materials from its samples (Sztrajman et al., 2021; Fan et al., 2022; Gokbudak et al., 2024). We show that isometric regularization can complement those methods to learn better latent space for BRDFs, resulting in better reconstruction quality.

We train neural BRDFs with the auto-decoder architecture by concatenating a latent variable $z$ with input directions, similar to Fan et al. (2022). We then optimize the latent codes to reconstruct the entire BRDF of unseen material from BRDF samples. To evaluate the results of BRDF reconstruction, we render a simple scene with the reconstructed BRDF with Mitsuba 3 renderer (Jakob et al., 2022). We compare the results of our method (IsoAD) with vanilla auto-decoder (AD) and auto-decoder with Lipschitz regularization (LipAD). We provide experimental details in Appendix C.2.

**Dataset.** We use the MERL dataset (Matusik et al., 2003), a common BRDF dataset measured from 100 real isotropic materials. We split the dataset into 80 materials for training and 20 materials for the test dataset. We train the model with various numbers of training data: $N = 20, 40, 60, 80$. We make five different datasets for $N = 20, 40, 60$ with a random choice of materials from the full training dataset ($N = 80$) and compute the average metrics for evaluating BRDF reconstruction.

**Results.** Figure 6 shows the reconstruction accuracy measured in PSNR and SSIM (Wang et al., 2004). Isometric regularization improves the BRDF reconstruction by a large margin for both metrics. The effect of regularization is prominent when the training data $N$ is greatly reduced to 20, maintaining robust results while the reconstruction results of AD drastically degrade. In particular, IsoAD trained with 20 materials shows higher PSNR and SSIM than the baselines trained with the full training dataset of 80 materials. This result demonstrates that isometric regularization enhances the generalization performance without compromising the fidelity of reconstruction. Qualitative results on BRDF reconstruction are shown in Figure 5. The quality of reconstructed materials aligns with what we expect from quantitative metrics.

## 5.3 NEURAL OPERATORS

The Deep Operator Network (DONet) (Lu et al., 2019), originally introduced for solving PDEs, is a standard architecture for constructing neural operators that map between functions. In this work, we demonstrate that incorporating isometric regularization can significantly improve the performance of operator learning. The neural operator is a regression task that aims to learn a mapping $\mathcal{G}$ from an input function $u : \mathcal{X} \to \mathbb{R}$ to an output function $o : \mathcal{Y} \to \mathbb{R}$, denoted by $\mathcal{G} : u(x) \mapsto o(y)$, using pairs of input-output functions. Specifically, we represent the input as a vectorized function $u_{\text{vec}} = (u(x_1), \ldots, u(x_M)) \in \mathcal{U}$ for a set of fixed points $\{x_k\}_{k=1}^M$. We denote the training dataset of size $N$ by $\{(u_{\text{vec},i}, o_i(y))\}_{i=1}^N$.

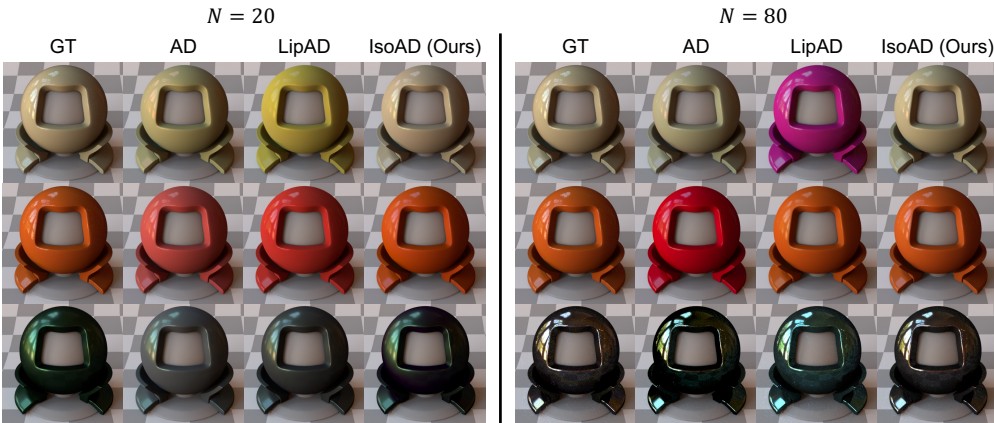

Figure 5: Qualitative results on BRDF reconstruction.

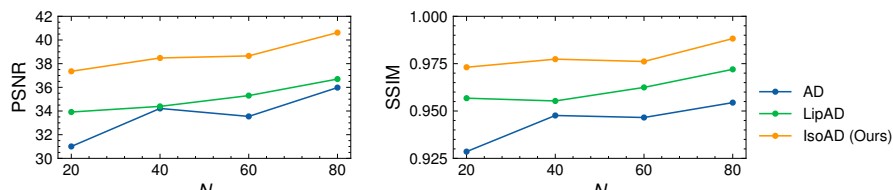

Figure 6: Average PSNR (left) and SSIM (right) on BRDF reconstruction.

Our model consists of two networks: an encoder $E : \mathcal{U} \to \mathcal{Z}$ and a decoder $F : \mathcal{Z} \times \mathcal{Y} \to \mathbb{R}$, where the output function is modeled as $o(y) = F(E(u_{\text{vec}}), y)$ for an input $u_{\text{vec}}$. When $u_{\text{vec}}$ is high-dimensional, $E$ encodes it into a low-dimensional vector that captures the essential information required to predict the output function. This structure allows $F$ to be interpreted as a latent variable model for functional representations, enabling the effective application of isometric regularization. We train $E$ and $F$ by minimizing $\sum_{i,j} \|o_i(y_j) - F(E(u_{\text{vec},i}), y_j)\|^2$ for a set of query points $\{y_j\}_{j=1}^{M'}$ and evaluate the regression performance on the test dataset. To assess the robustness of the model, we introduce varying levels of noise to the output functions in the training dataset, while keeping the input functions clean, simulating real-world uncertainties such as measurement errors or numerical approximation inaccuracies. We compare the performance of three models: the unregularized DONet, the model with Lipschitz regularization (LipDONet), and the one with isometric regularization (IsoDONet).

**Dataset.** This study focuses on two types of PDE datasets: the reaction-diffusion equation, as discussed in Yang et al. (2022), and the Darcy flow problem, based on Lu et al. (2022). The reaction-diffusion equation describes how a solution $u(t, x)$ evolves over time and space under the effects of diffusion and reaction forces:

$$\frac{\partial}{\partial t} u(t, x) = \nu \frac{\partial^2}{\partial x^2} u(t, x) + k u^2 + f(x), \quad (t, x) \in [0, 1] \times [0, 1] \tag{10}$$

with some initial and boundary conditions. The operator we aim to learn is a mapping $\mathcal{G} : f(x) \mapsto u(t, x)$, where $f(x)$ represents the forcing term.

The Darcy flow problem models fluid flow through a porous medium, of which steady state on a unit square is given by:

$$\nabla \cdot (a(x, y) \nabla u(x, y)) = f(x, y), \quad (x, y) \in [0, 1]^2 \tag{11}$$

with some boundary conditions. The operator we aim to learn is a mapping $\mathcal{G} : a(x, y) \mapsto u(x, y)$ where $a(x, y)$ is the diffusion coefficient and $u(x, y)$ is fluid density, while assuming a fixed external force $f(x, y)$. These datasets are generated by solving the PDEs; details regarding the datasets can be found in Appendix C.3.

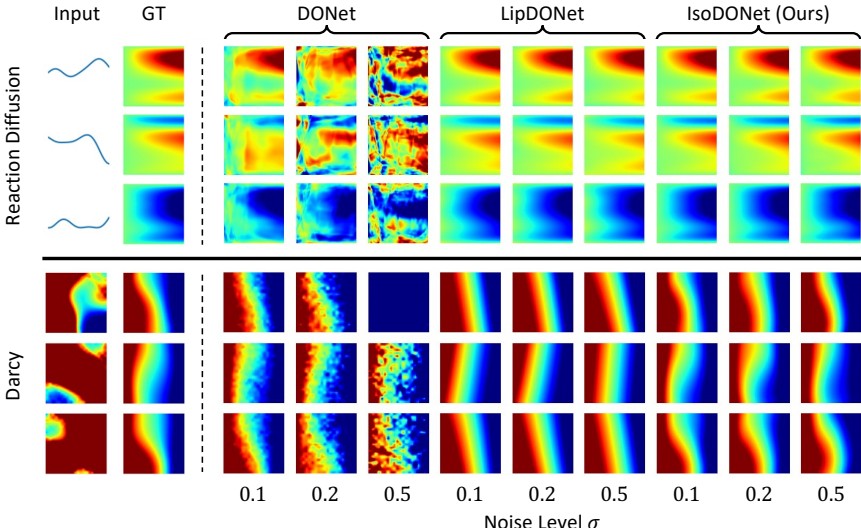

Figure 7: Qualitative results on neural operator. In the output images of reaction-diffusion, the horizontal axis represents $t$ and the vertical axis represents $x$.

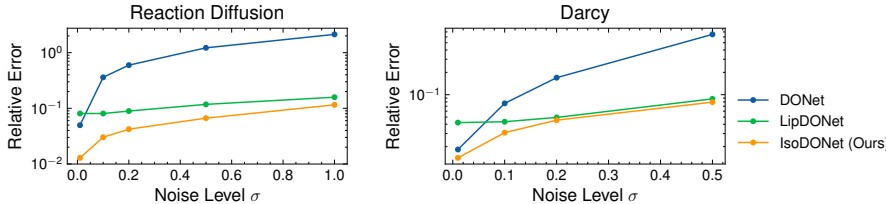

Figure 8: Relative errors as functions of the noise levels added to the training output functions.

**Results.** Figure 8 shows the relative errors (or root mean square percentage errors) measured on clean test datasets as functions of the noise levels added to the training output functions. First, it can be observed that the errors increase for all three methods as the noise level increases. Importantly, for IsoDONet, the rate of increase is much smaller compared to the other methods, and it consistently shows lower errors. This demonstrates that isometric regularization is a highly effective regularization technique. Figure 7 shows some prediction results given input functions from the trained operator models that were trained with corrupted training datasets. DONet produces highly distorted predictions, severely overfitting to the noise in the data, and as expected, the distortion becomes greater as the noise increases. LipDONet, while performing much better than DONet, tends to overly regularize the output function in the spatio-temporal domain, especially as observed in the Darcy flow problem, when compared to the ground truth (GT) output function on the left. It can be observed that IsoDONet produces the best qualitative results, predicting the output function with relatively less distortion and without overfitting to the noise.

## 6 CONCLUSION

In this work, we have proposed the isometric regularization for manifolds on infinite-dimensional functional data space. We define inner products of functions with probabilistic weights and adapt Riemannian manifold learning within the formulation of latent variable models of INR. With extensive experiments on various modalities of functional data, we have confirmed that our isometric regularization can balance data fidelity and generalization, demonstrating improved performance on downstream tasks including interpolation and reconstruction. However, there are still arbitrary choices in the process of mapping the infinite dimension into the latent space, such as the function-dependent inner product $p(x; F_z)$ in the infinite-dimensional function data or the dimension of the latent variable $\mathcal{Z}$. An interesting future direction is to design a more systematic formulation suitable for downstream applications according to the data characteristics.

ACKNOWLEDGMENTS

We would like to thank Clément Jambon for his valuable help with Mitsuba 3 renderer. This work was supported by Institute for Information & Communications Technology Planning & Evaluation (IITP) grant funded by the Korea government (MSIT) [No.2021-0-01341 (Artificial Intelligence Graduate School Program (Chung-Ang University)), No.RS-2021-II211343 (Artificial Intelligence Graduate School Program (Seoul National University)), No.2021-0-02068 (Artificial Intelligence Innovation Hub)], CAINS through the Center for AI and Natural Sciences at Korea Institute for Advanced Study (KIAS), and by KIAS Individual Grant No. AP092701.

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

APPENDIX

## A    PROOF OF RELAXED DISTORTION MEASURE

Denote $J(x, z) = \frac{\partial F}{\partial z}(x, z) \in \mathbb{R}^{1 \times m}$ for $z \in \mathbb{R}^m$ and $x \in \mathcal{X}$. In this section, we show that $F(x, z)$ is a scaled isometric function generative model in the support of $P_\mathcal{Z}$, i.e., for some positive scalar $c$,

$$H(z) := \int J^T(x, z) J(x, z) p(x; F_z) \, dx = cI \quad \text{for all} \quad z \in \text{supp}(P_\mathcal{Z}) \tag{12}$$

if and only if $F$ is any minimizer of

$$\frac{\mathbb{E}_{z \sim P_\mathcal{Z}}[\text{Tr}(H^2(z))]}{\mathbb{E}_{z \sim P_\mathcal{Z}}[\text{Tr}(H(z))]^2}. \tag{13}$$

This proof is based on the Proposition 2 in Lee et al. (2022b). Denoting the eigenvalues of $H(z)$ by $\lambda_i(z), i = 1, \ldots, m$, we can re-write the above equation as follows:

$$\frac{\mathbb{E}_{z \sim P_\mathcal{Z}}[\text{Tr}(H^2(z))]}{\mathbb{E}_{z \sim P_\mathcal{Z}}[\text{Tr}(H(z))]^2} = \frac{\mathbb{E}_{z \sim P_\mathcal{Z}}[\sum_{i=1}^m \lambda_i^2(z)]}{\mathbb{E}_{z \sim P_\mathcal{Z}}[\sum_{i=1}^m \lambda_i(z)]^2}$$

$$= \mathbb{E}_{z \sim P_\mathcal{Z}}[\sum_{i=1}^m \Big(\frac{\lambda_i(z)}{\mathbb{E}_{z \sim P_\mathcal{Z}}[\sum_{i=1}^m \lambda_i(z)]}\Big)^2]$$

$$= \mathbb{E}_{z \sim P_\mathcal{Z}}[\sum_{i=1}^m \Big(\Big(\frac{\lambda_i(z)}{\mathbb{E}_{z \sim P_\mathcal{Z}}[\sum_{i=1}^m \lambda_i(z)]}\Big)^2 - \frac{2}{m}\Big(\frac{\lambda_i(z)}{\mathbb{E}_{z \sim P_\mathcal{Z}}[\sum_{i=1}^m \lambda_i(z)]}\Big) + \frac{1}{m^2}\Big)] + \frac{2}{m} - \frac{m}{m^2}$$

$$= \mathbb{E}_{z \sim P_\mathcal{Z}}[\sum_{i=1}^m \Big(\frac{\lambda_i(z)}{\mathbb{E}_{z \sim P_\mathcal{Z}}[\sum_{i=1}^m \lambda_i(z)]} - \frac{1}{m}\Big)^2] + \frac{1}{m}. \tag{14}$$

Therefore, Equation (13) is minimal if and only if

$$\frac{\lambda_i(z)}{\mathbb{E}_{z \sim P_\mathcal{Z}}[\sum_{i=1}^m \lambda_i(z)]} - \frac{1}{m} = 0 \quad \text{for all} \quad z \in \text{supp}(P_\mathcal{Z}). \tag{15}$$

Now, we prove both directions. ($\implies$) suppose $H(z) = cI$ at all $z \in \text{supp}(P_\mathcal{Z})$ for some positive scalar $c$, then $\lambda_i(z) = c$ for all $z \in \text{supp}(P_\mathcal{Z})$ and $i$. Thus Equation (15) holds true and Equation (13) is minimal. ($\impliedby$) suppose Equation (15) holds true, then $\lambda_i(z) = \frac{1}{m}\mathbb{E}_{z \sim P_\mathcal{Z}}[\sum_{i=1}^m \lambda_i(z)]$ that is some positive constant $c$ for all $z \in \text{supp}(P_\mathcal{Z})$, which ends the proof.

## B    APPROXIMATE COMPUTATION OF RELAXED DISTORTION MEASURE

In this section, we explain details for approximately computing Equation (13) during the training of $F(x, z)$. To further elaborate on the computation of Jacobian-vector and vector-Jacobian products, first consider the $\text{Tr}(\mathbb{E}_x[H(z)]) = \text{Tr}(\mathbb{E}_x[J^T J])$ term in the denominator. Using the Hutchinson's estimator, this trace term can be approximated with $\frac{1}{N}\sum_{k=1}^N (v_k^T \mathbb{E}_x[J^T J]v_k)$ for $v_k \sim \mathcal{N}(0, I_m)$ – where $I_m$ is an $m \times m$ identity matrix. Since $v_k^T \mathbb{E}_x[J^T J]v_k = \mathbb{E}_x(v_k^T J^T J v_k)$, we need to compute $J(x, z)v_k = \frac{\partial F}{\partial z}v_k$.

Removing the averaging term of trace estimator for simplicity, the denominator becomes

$$\mathbb{E}_{z \sim P_\mathcal{Z}}[\text{Tr}(H(z))]^2 = \mathbb{E}_{z \sim P_\mathcal{Z}, x \sim p(x; F_z)}[v_k^T J^T J v_k]^2$$

$$= \mathbb{E}_{z \sim P_\mathcal{Z}, x \sim p(x; F_z)}[\|Jv_k\|^2]^2. \tag{16}$$

Since $F$ is a function of $(x, z)$, the conventional Jacobian-vector product implemented via automatic differentiation operates on a vector of size $(x, z)$. Therefore, to compute $Jv_k$, we augment the vector $v$ with zero vector of size $x$, denoted by $\vec{0}$, and compute the $J(x, z)v_k$ as $\frac{\partial F}{\partial (x, z)}(\vec{0}, v)$. Similarly, if we examine the numerator,

$$\mathbb{E}_{z \sim P_\mathcal{Z}}[\text{Tr}(H^2(z))] = \mathbb{E}_{z \sim P_\mathcal{Z}, x \sim p(x; F_z)}[v_k^T J^T J J^T J v_k]$$

$$= \mathbb{E}_{z \sim P_\mathcal{Z}, x \sim p(x; F_z)}[\|J^T J v_k\|^2]. \tag{17}$$

Let the output of Jacobian-vector product $Jv_k = w \in V$, we need to compute $J^T w = \frac{\partial F}{\partial z}^T w$. We compute this by taking the last $m$-th components from the conventional vector-Jacobian product $w^T \frac{\partial F}{\partial(x,z)} \in \mathbb{R}^{\dim(\mathcal{X})+m}$.

We note that if $F$ has certain specialized architectures, such as a neural network where parameters are not shared between $z$ and $x$, more efficient implementations of these products may be feasible without the need for zero vector padding.

## C  EXPERIMENTAL DETAILS

### C.1  NEURAL SDFs

#### C.1.1  TOY EXAMPLES ON SIMILAR SHAPES

**Training.**  We train a multi-layer perceptron (MLP) with fully connected layers with a sequence of nodes $(4, 128, 128, 128, 128, 1)$ with ReLU activation functions. The input is the 2-dimensional latent code concatenated with 2D spatial point and the output is the signed distance value of the corresponding shape. We initialize the latent codes $z \sim \mathcal{N}(0, 0.1^2)$ and optimize the network parameters and the latent codes jointly with MSE loss between the output of the network and the ground truth SDF samples. We use Adam (Kingma, 2014) with a learning rate of 1e-4 for network parameters and 1e-3 for the latent codes. We uniformly sample 512 points on the unit square for input coordinate samples for isometric regularization. We perform parameter sweeping for the weight of isometric regularization on a log scale and report the best case for each model regarding test accuracy on linear interpolation in Table 3.

Table 3: Weights of isometric regularization on toy examples.

| $N$ | Circle | Square | Triangle |
|---|---|---|---|
| 5 | 0.001 | 0.001 | 0.0001 |
| 10 | 0.001 | 1e-5 | 0.0001 |
| 20 | 0.001 | 1e-5 | 0.0001 |

#### C.1.2  2D SURFACE RECONSTRUCTION

**Training.**  We train neural SDFs with the MNIST dataset for 2D surface reconstruction, similar to Appendix C.1.1. We use fully connected neural network with a sequence of nodes $(4, 256, 256, 256, 256, 256, 1)$ with ReLU activation functions. The input is the concatenation of the 8-dimensional latent code and the 2D spatial point. We initialize the latent codes $z \sim \mathcal{N}(0, 0.1^2)$ and optimize the network parameters and the latent codes jointly with MSE loss between the output of the network and the ground truth SDF samples. We use Adam with a learning rate of 1e-4 for network parameters and 1e-3 for the latent codes. We uniformly sample 4096 points on the unit square for input coordinate samples for isometric regularization. We perform parameter sweeping for the weight of each regularization (Lipschitz and ours) on a log scale and report the best case for each model regarding the accuracy of test time optimization in Table 4.

Table 4: Weights of regularization terms on 2D surface reconstruction.

| $N$ | Lipschitz | Isometric (Ours) |
|---|---|---|
| 100 | 1e-7 | 0.01 |
| 500 | 1e-7 | 0.001 |

**Reconstruction.**  We reconstruct the 2D surface given 256 points randomly sampled from the zero-level set of digits from the test dataset by optimizing the latent code during test time. The test dataset consists of 100 images for each digit. We randomly initialize the latent code $z \sim \mathcal{N}(0, 1^2)$ and optimize the latent code minimizing the L1 norm of the output, as the SDF values of the input points

should be zero. We use Adam with a learning rate of 1e-4 and iterate until the loss converges. As the initialization fairly affects the reconstruction results, we repeat the optimization process twice with different initializations for each test data and report the result with a lower loss convergence.

### C.1.3 3D SURFACE RECONSTRUCTION

**Training.** We train autoencoder for 3D surface reconstruction, following Liu et al. (2022). The encoder is a Pointnet (Qi et al., 2017), which gets 3D spatial points for input. The encoder consists of two blocks of MLP with fully connected layers. The first block has a sequence of nodes $(3, 256, 512)$. After the hidden layer of 256 neurons, the input passes through the layer normalization and tanh activation before the 512-dimensional output feature vector. As the input points are $n \times 3$ matrix, the output of the first block is the $n \times 512$ feature matrix. Then, we obtain a global feature vector $z_{\text{global}} \in \mathbb{R}^{512}$ by max-pooling the feature matrix. We expand the global feature vector by repeating $n$ times and concatenate it with the feature matrix (output of the first block). The second block has the sequence of nodes $(1024, 512, 256)$. Same as the first block, we pass through the layer normalization and tanh activation after the hidden layer. We then perform another max-pooling to get the final global feature vector. We apply the sigmoid function to the final global feature vector to get the latent code lying between 0 and 1.

The decoder follows the architecture of DeepSDF (Park et al., 2019). The input of the decoder is the latent code concatenated with the spatial query point. The decoder is a fully connected neural network with a sequence of nodes $(259, 1024, 1024, 1024, 512, 256, 128, 1)$, each hidden layer followed by layer normalization and leaky ReLU activation. Also, we perform dropout with probability 0.2 for each hidden layer and skip connection by concatenating the input latent vector to the output of the fourth hidden layer.

For the training dataset, we sample SDF samples following Park et al. (2019). For input point clouds to the encoder, we randomly sample points on the surface. We use L1 loss between the predicted SDF from the decoder and the ground truth SDF. We optimize the network parameters with Adam with a learning rate of 5e-3 for 2000 epochs. We adjust the learning rate by half per 500 epochs.

As we train the truncated SDF (TSDF) with a clamping distance of 0.1, we sample spatial points near the surface for isometric regularization. Please refer to Appendix D.1 for the choice of sampling strategy for $p(x; F_z)$. Specifically, we randomly sample 4096 points with a distance smaller than 0.1. We perform parameter sweeping for the weight of each regularization (Lipschitz and ours) on a log scale and report the best case for each model regarding the accuracy of test time optimization in Table 7.

Table 5: Weights of regularization terms on 3D reconstruction

| Dataset | Lipschitz | | | | | Isometric (Ours) | | | | |
|---|---|---|---|---|---|---|---|---|---|---|
| | 1 | 2 | 3 | 4 | 5 | 1 | 2 | 3 | 4 | 5 |
| $N = 5$ | 1e-9 | 1e-10 | 1e-11 | 1e-9 | 1e-11 | 0.0001 | 1e-5 | 0.0001 | 0.0001 | 0.0001 |
| $N = 10$ | 1e-11 | 1e-9 | 1e-11 | 1e-11 | 1e-11 | 0.0001 | 0.001 | 0.001 | 0.001 | 1e-5 |

**Reconstruction.** We reconstruct the full 3D surface given partial observations of a zero-level set by optimizing the latent code during test time. The latent code is initialized by passing forward through the encoder. Then we optimize the latent code parameters (parameter before applying the sigmoid function) by minimizing the loss function. Our loss function for the test time optimization is as follows: $\mathcal{L} = \mathcal{L}_1 + \lambda_{\text{eikonal}}\mathcal{L}_{\text{eikonal}}$, where $\mathcal{L}_1$ is the L1 norm of the output of the network, $\mathcal{L}_{\text{eikonal}}$ is the eikonal term, and $\lambda_{\text{eikonal}}$ is the weight for eikonal term. $\mathcal{L}_{\text{eikonal}}$ is the regularization term to force the norm of the gradient of the output to be 1 (Gropp et al., 2020). We use Adam with a learning rate of 1e-4 and iterate until the loss $\mathcal{L}$ converges. We set $\lambda_{\text{eikonal}} = 0.01$.

### C.2 NEURAL BRDFS

**Training.** We train neural BRDFs with the auto-decoder architecture similar to Fan et al. (2022). However, we simplify the network architecture to MLP with fully connected layers, as we use a

BRDF dataset from simple isotropic materials instead of complex materials with multiple layers. The network consists of 8 hidden layers of 256 neurons with GELU activation (Hendrycks & Gimpel, 2016). We add a final exponential layer to the MLP's output as BRDF values have a high dynamic range. The network input is a 12 dimension latent vector $z$ and 6 dimension input direction vectors. The network output is a 1-dimensional scalar, the BRDF value of each RGB color channel. Thus, each BRDF is represented by three latent vectors optimized for each RGB channel.

Following Sztrajman et al. (2021), we parametrize the input directions as the Cartesian vector $\mathbf{h}$ and $\mathbf{d}$ in the Rusinkiewicz parametrization (Rusinkiewicz, 1998). As BRDF values have a high dynamic range, we use logarithmic loss $\mathcal{L} = |\log(1 + f_r^{\text{gt}}) - \log(1 + f_r^{\text{pred}})|$ for training, similar to Sztrajman et al. (2021).

We randomly initialize the latent vectors $z \sim \mathcal{N}(0, 0.1^2)$. Then, we jointly optimize the latent vectors and network parameters with Adam by minimizing the training loss $\mathcal{L}$. The learning rate is 5e-4 for the network parameters and 1e-4 for the latent codes. We trained the models for 200 epochs. We decreased the learning rates by half after 100 epoch.

For input coordinate samples for isometric regularization, we uniformly sample from the angle space of input directions ($\theta_{\mathbf{h}}, \phi_{\mathbf{h}}$, and $\phi_{\mathbf{d}}$). We perform parameter sweeping for the weight of each regularization (Lipschitz and ours) on a log scale and report the best case for each model regarding the accuracy of test time optimization in Table 6.

Table 6: Weights of regularization terms on BRDF reconstruction.

| Dataset | Lipschitz | | | | | Isometric (Ours) | | | | |
|---|---|---|---|---|---|---|---|---|---|---|
| | 1 | 2 | 3 | 4 | 5 | 1 | 2 | 3 | 4 | 5 |
| $N = 20$ | 1e-8 | 1e-9 | 1e-8 | 1e-8 | 1e-8 | 0.0001 | 0.001 | 1e-5 | 1e-5 | 1e-5 |
| $N = 40$ | 1e-10 | 1e-8 | 1e-9 | 1e-10 | 1e-10 | 1e-5 | 1e-5 | 0.0001 | 0.001 | 0.001 |
| $N = 60$ | 1e-11 | 1e-11 | 1e-10 | 1e-10 | 1e-9 | 0.001 | 0.001 | 0.0001 | 1e-5 | 0.0001 |
| $N = 80$ | 1e-9 | N/A | N/A | N/A | N/A | 1e-5 | N/A | N/A | N/A | N/A |

**Reconstruction.** We reconstruct the BRDF from samples by optimizing the latent codes during test time. We randomly initialize the latent codes $z \sim \mathcal{N}(0, 0.1^2)$. Then we optimize the latent codes by minimizing the logarithmic loss between the network output and the BRDF samples. We use Adam and set the learning rate as 1e-4. As the initialization fairly affects the reconstruction results, we repeat the optimization 3 times with different initializations and report the best regarding the convergence loss.

## C.3 NEURAL OPERATORS

**Datasets.** First, we consider the reaction-diffusion equation covered in Yang et al. (2022). The reaction-diffusion equation is written as

$$
\begin{aligned}
\frac{\partial}{\partial t}u(t,x) &= \nu\frac{\partial^2}{\partial x^2}u(t,x) + ku^2 + f(x), & (t,x) \in [0,1] \times [0,1], \\
u(t=0,x) &= 0, & x \in [0,1], \\
u(t,x=0) = u(t,x=1) &= 0, & t \in [0,1],
\end{aligned}
\tag{18}
$$

where $\nu$ is the diffusion coefficient and $k$ is the reaction rate. We set $\nu = 0.01$ and $k = 0.01$. We then consider the operator $\mathcal{G} : f(x) \mapsto u(t,x)$ where $f(x)$ is generated from a Gaussian process prior with an exponential quadratic kernel using a length scale parameter $l = 0.2$. For the projection of input and output functions, the time domain $t \in [0,1]$ and spatial domain $x \in [0,1]$ are uniformly discretized into 100 points each. Standard Gaussian noise with varying standard deviations is added to the training output function to generate corrupted datasets.

For Darcy flow, we regenerated the data using the same procedure as in Lu et al. (2022). This problem involves a diffusion equation with an external force, modeling fluid flow through a porous medium. The steady state on a unit square is given by:

$$
\begin{cases}
\nabla \cdot (a(x,y)\nabla u(x,y)) = f(x,y), & (x,y) \in [0,1]^2 \\
u(x,y) = 0, & (x,y) \in \partial(0,1)^2,
\end{cases}
\tag{19}
$$

where $u(x, y)$ is the fluid density, $a(x, y)$ is the diffusion coefficient, and $f(x, y)$ is the external force. Our goal is to learn the mapping $\mathcal{G} : a(x, y) \mapsto u(x, y)$ with a fixed external force $f(x, y) = 1$. The input $a(x, y)$ and the output $u(x, y)$ are uniformly discretized to a resolution of $20 \times 20$. Standard Gaussian noise with varying standard deviations is added to the training output function to generate corrupted datasets.

**Training.** We train neural operators consisting of $E : \mathcal{U} \to \mathcal{Z}$ and $F : \mathcal{Z} \times \mathcal{Y} \to \mathbb{R}$, where $F$ is further composed of branch and trunk networks, as adopted from Lu et al. (2019):

$$F(z, y) := \sum_{b=1}^{N_b} B_b(y) T_b(z), \qquad (20)$$

where $B_b : \mathcal{Y} \to \mathbb{R}$ and $T_b : \mathcal{Z} \to \mathbb{R}$ for $b = 1, \ldots, N_b$. The vectors $B = (B_1, \ldots, B_{N_b})$ and $T = (T_1, \ldots, T_{N_b})$ are referred to as the branch and trunk networks, respectively. For the Reaction-Diffusion problem, $E$ is set as the identity map, i.e., $z = u_{\text{vec}}$. The map $B$ is a fully connected neural network with a sequence of nodes $(100, 256, 256, 256, 256, 100)$ and GELU activation functions, while $T$ is a fully connected neural network with a sequence of nodes $(2, 256, 256, 256, 256, 100)$ and GELU activation functions, where $N_b = 100$. For Darcy Flow, $E$ is set as the resnet18 followed by the linear layer that maps 512-dimensional vector to 32-dimensional vector $z$. The map $B$ is a fully connected neural network with a sequence of nodes $(32, 1024, 1024, 1024, 1024, 100)$ and GELU activation functions, while $T$ is a fully connected neural network with a sequence of nodes $(2, 256, 256, 256, 256, 100)$ and GELU activation functions, where $N_b = 100$.

The learning rate is set to 1e-4. We trained the models for 200,000 epochs with a batch size of 1,000 for the reaction-diffusion datasets and for 55,000 epochs with a batch size of 200 for the Darcy flow datasets. We perform a parameter sweep for the weight of each regularization (Lipschitz and ours) for each corrupted dataset separately and report the best case for each model based on the test set error.

Table 7: Weights of regularization terms on Neural Operator leraning.

| Noise level | Lipschitz | | | | | Isometric (Ours) | | | | |
|---|---|---|---|---|---|---|---|---|---|---|
| | 0.01 | 0.1 | 0.2 | 0.5 | 1.0 | 0.01 | 0.1 | 0.2 | 0.5 | 1.0 |
| Reaction Diffusion | 0.0001 | 0.0001 | 0.0001 | 0.0001 | 0.001 | 1 | 1 | 10 | 30 | 100 |
| Darcy Flow | 0.0001 | 0.0001 | 0.0001 | 0.001 | N/A | 0.1 | 10 | 30 | 100 | N/A |

## D  ADDITIONAL EXPERIMENTAL RESULTS

### D.1  ABLATION ON $p(x; F_z)$

In this section, we perform an ablation study on the probability density $p(x; F_z)$ with neural SDFs. We compare two sampling strategies for $p(x; F_z)$: uniform sampling and near-surface sampling. We train the models with SDF samples from the toy example on five similar circles with radius 0.1 to 0.5. To consider various data characteristics, we also train TSDF samples by clamping the distance of SDF samples with $\pm 0.1$. We follow the experimental details of toy examples on similar shapes described in Appendix C.1.1. We visually analyze the effect of $p(x; F_z)$ on the latent space according to the training data (SDF & TSDF).

Figure 9 shows the latent space of neural SDFs trained with SDF and TSDF samples from five circles. We show decoded SDFs&TSDFs and L1 errors of linear interpolants in Figure 10 and Figure 11. While other cases show nearly the same results expected with isometric regularization, showing straight equidistant latent space, isometric regularization with uniform sampling for TSDF straightens the latent space with "non-equidistant" latent codes. Specifically, the distance between the optimized latent codes increases as the radius increases. These results are due to the differences in data characteristics between SDF and TSDF. TSDF values from the points far from the surface than 0.1 are constant with the change of the latent code. As samples from those points do not affect the computation of Equation (12), shapes with small surface areas should be closer on the

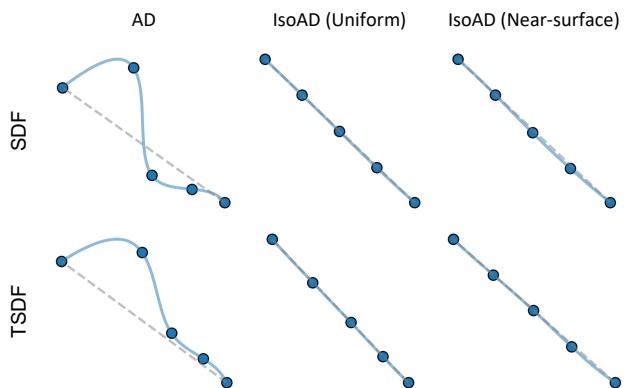

Figure 9: Latent space trained with SDF & TSDF from 5 circles.

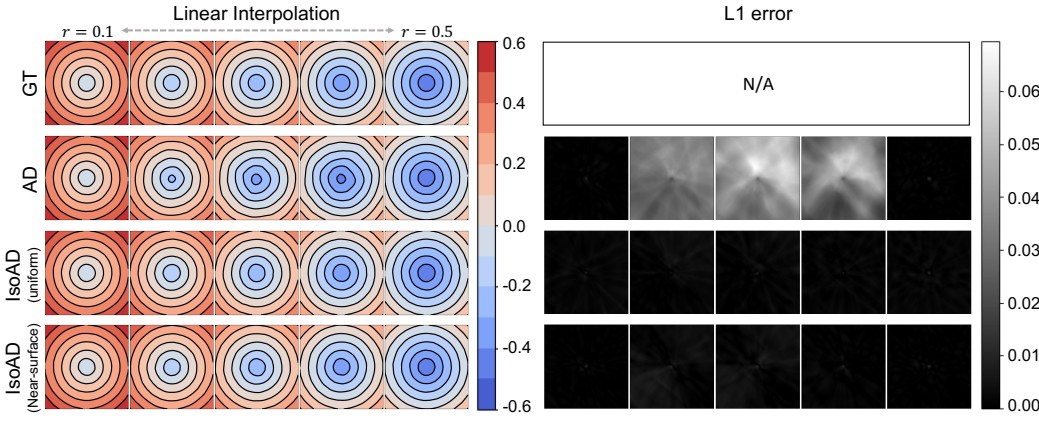

Figure 10: Linear interpolation on latent space trained with SDF.

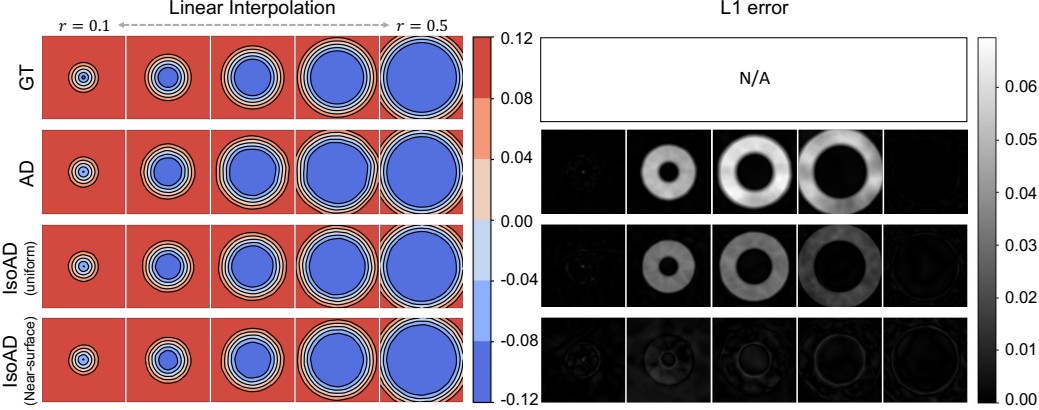

Figure 11: Linear interpolation on latent space trained with TSDF.

latent manifold than those with larger surface areas when uniform sampling is used for training TSDF. Therefore, concentrating the samples near the surface for isometric regularization achieves the same expected results for both cases trained with SDF and TSDF. We note that there could be numerous choices of $p(x; F_z)$ depending on the nature of the data and the expected effect of isometric regularization.

## D.2 REGULARIZATION WEIGHT ANALYSIS

In this section, we provide an analysis on weights of regularization terms. The scale of the relaxed distortion measure varies across datasets, necessitating weighting factor tuning for each dataset. Empirically, we start by setting the weighting factor to match the scale of the isometric regularization term similar to the original training loss term and then change the weighting factor with parameter sweeping. Tables 8, 9, and 10 show the results with various weighting factors on the toy dataset, MNIST dataset, and MERL dataset, respectively. The results of $N = 20, 40$ and $60$ in Table 10 are the average PSNR and SSIM over five datasets with the same weights on regularization term. When the weights on the isometric regularization are too large, the network may struggle to fit the training data, resulting in reduced performance compared to results without regularization. However, our method consistently outperforms those without isometric regularization across most weighting settings, demonstrating sufficient robustness to the choice of the weighting factor.

Table 8: Results of regularization weight analysis on toy dataset ($N = 5$).

|  | AD | IsoAD (Ours) | | | | |
|---|---|---|---|---|---|---|
| weight | N.A. | 0.1 | 0.01 | 0.001 | 0.0001 | 1e-5 |
| L1 error | 36.5885 | 4.3318 | **2.6013** | 2.7497 | 9.7079 | 20.5224 |

Table 9: Results of regularization weight analysis on MNIST dataset ($N = 100$).

|  | DeepSDF | LipDeepSDF | | | IsoDeepSDF (Ours) | | |
|---|---|---|---|---|---|---|---|
| weight | N.A. | 1e-06 | 1e-07 | 1e-08 | 0.01 | 0.001 | 0.0001 |
| Average L1 error | 0.03656 | 0.03511 | 0.03507 | 0.03854 | **0.02829** | 0.03302 | 0.03032 |
| Median L1 error | 0.03551 | 0.03401 | 0.03413 | 0.03755 | **0.02654** | 0.03133 | 0.02933 |

Table 10: Results of regularization weight analysis on MERL dataset.

|  |  | AD | LipAD | | | | IsoAD (Ours) | | |
|---|---|---|---|---|---|---|---|---|---|
|  | weight | N.A. | 1e-8 | 1e-9 | 1e-10 | 1e-11 | 0.001 | 0.0001 | 1e-5 |
| $N = 80$ | PSNR | 35.9776 | 35.5008 | 36.6982 | 33.9015 | 35.4429 | 32.7568 | 39.0047 | 40.6211 |
|  | SSIM | 0.9544 | 0.9402 | 0.9720 | 0.9582 | 0.9326 | 0.9426 | 0.9767 | 0.9882 |
| $N = 60$ | PSNR | 33.5426 | 32.5683 | 32.6521 | 33.2074 | 34.0972 | 35.8485 | 37.4657 | 35.3189 |
|  | SSIM | 0.9466 | 0.9483 | 0.9504 | 0.9459 | 0.9561 | 0.9585 | 0.9691 | 0.9557 |
| $N = 40$ | PSNR | 33.3672 | 34.4997 | 31.8454 | 33.8662 | 31.4173 | 36.4007 | 34.2912 | 35.9980 |
|  | SSIM | 0.9383 | 0.9613 | 0.9381 | 0.9591 | 0.9237 | 0.9635 | 0.9502 | 0.9590 |
| $N = 20$ | PSNR | 31.0048 | 33.0056 | 31.9782 | 31.0097 | 29.4476 | 33.7202 | 34.9357 | 34.6490 |
|  | SSIM | 0.9286 | 0.9450 | 0.9336 | 0.9253 | 0.9155 | 0.9429 | 0.9550 | 0.9557 |

## D.3 COMPUTATIONAL TIME

In this section, we provide the per-epoch runtime of AD, LipAD, and IsoAD during the training of neural SDFs with toy examples and the MNIST dataset. We use $N = 5$ circles for toy examples and $N = 100$ for the MNIST dataset with the same experimental settings in Appendix C.1. We use a single NVIDIA GeForce RTX 2080 Ti GPU for training. Table 11 shows the per-epoch runtime.

## D.4 REGULARIZATION INTERVAL ANALYSIS

While our Algorithm 1 avoids costly Jacobian computations, our method still requires considerable additional computation regarding training batch size and number of samples on $\mathcal{X}$. One simple yet effective way to accelerate training is to add the isometric regularization term at specific intervals

Table 11: Averages and standard deviations of the per-epoch runtimes (s) during training for each dataset.

| Dataset | Toy Example | | MNIST | | |
|---|---|---|---|---|---|
| Models | AD | IsoAD (Ours) | DeepSDF | LipDeepSDF | IsoDeepSDF (Ours) |
| per-epoch runtime (s) | 0.0049 ($\pm$ 0.002) | 0.0152 ($\pm$ 0.0064) | 0.8793 ($\pm$ 0.0543) | 0.9256 ($\pm$ 0.042) | 2.0819 ($\pm$ 0.0735) |

rather than every epoch. To show the robustness of our regularization method to the regularization interval, we conduct additional experiments on neural SDFs and neural BRDFs. We use the toy example dataset with circles ($N = 5$), MNIST dataset ($N = 100$), and MERL dataset ($N = 20$). We use the same experimental settings in Appendix C.1 for neural SDFs and Appendix C.2 for neural BRDFs. Tables 12, 13, and 14 show the results of regularization interval analysis on the toy example dataset, MNIST dataset, and MERL dataset, respectively.

Table 12: Results of regularization interval analysis on toy example dataset (circles: $N = 5$). We show average L1 errors ($\times 10^{-3}$) on linear interpolation.

| | AD | IsoAD (Ours) | | | |
|---|---|---|---|---|---|
| interval | N.A. | 1 | 2 | 5 | 10 |
| weight | N.A. | 0.01 | 0.01 | 0.01 | 0.01 |
| L1 error | 36.5885 | 2.6014 | **1.8571** | 3.2573 | 3.1487 |

Table 13: Results of regularization interval analysis on MNIST dataset ($N = 100$). We show average and median L1 errors on reconstruction from zero-level sets.

| | DeepSDF | LipDeepSDF | IsoDeepSDF (Ours) | | | |
|---|---|---|---|---|---|---|
| interval | N.A. | 1 | 1 | 2 | 5 | 10 |
| weight | N.A. | 1.00E-07 | 0.01 | 0.01 | 0.05 | 0.05 |
| Average L1 error | 0.0366 | 0.0351 | **0.0283** | 0.0330 | 0.0335 | 0.0359 |
| Median L1 error | 0.0355 | 0.0341 | **0.0265** | 0.0321 | 0.0322 | 0.0338 |

Table 14: Results of regularization interval analysis on MERL dataset ($N = 20$). We show average PSNR and SSIM over five datasets of $N = 20$ on BRDF reconstruction.

| | AD | LipAD | IsoAD (Ours) | | | |
|---|---|---|---|---|---|---|
| interval | N.A. | 1 | 1 | 2 | 5 | 10 |
| PSNR | 31.0048 | 33.9142 | **37.3545** | 36.4126 | 36.3188 | 36.0928 |
| SSIM | 0.9286 | 0.9568 | **0.9731** | 0.9498 | 0.9514 | 0.9471 |

## D.5 ADDITIONAL RESULTS

### D.5.1 TOY EXAMPLES ON SIMILAR SHAPES

We show additional results of neural SDFs trained with simple 2D shapes. Figure 12 shows the latent space of AD and IsoAD each trained with five squares and five triangles. Figure 13 and Figure 14 show generated intermediate shapes by linearly interpolating the latent space given the smallest and the biggest shapes of each model.

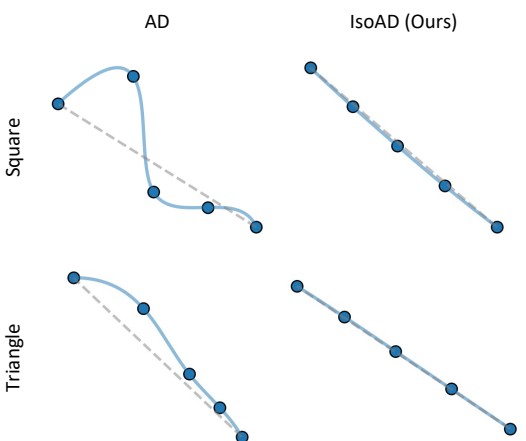

Figure 12: Latent space trained with five squares & triangles.

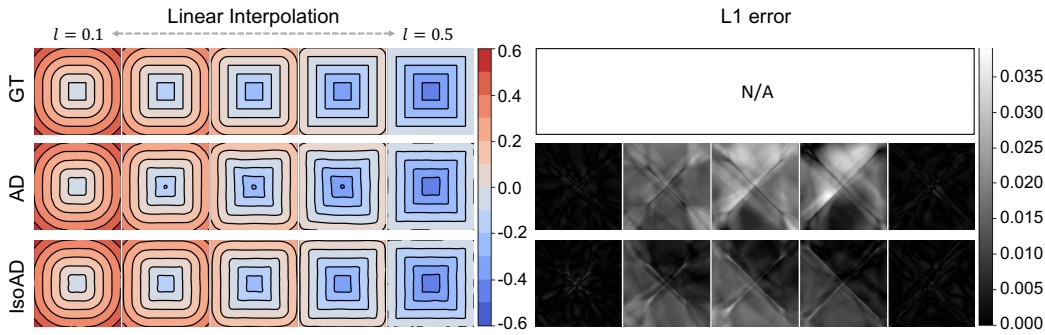

Figure 13: Linear interpolation on latent space trained with five squares.

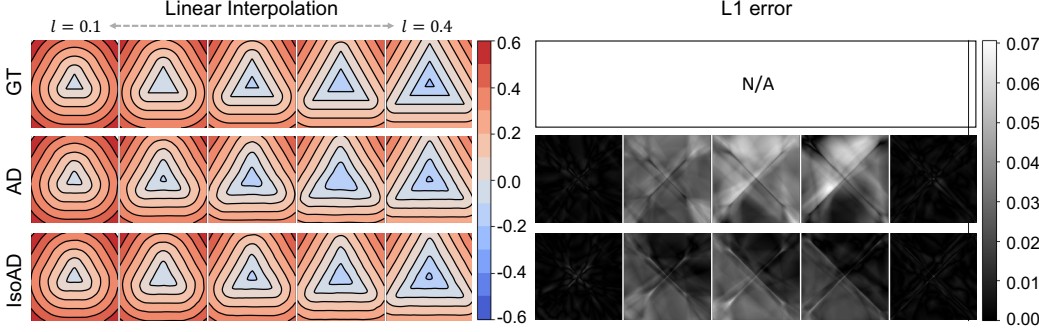

Figure 14: Linear interpolation on latent space trained with five triangles.

## D.6 SURFACE RECONSTRUCTION

We show additional qualitative results of surface reconstruction on MNIST and ShapeNet datasets. Figure 15 shows the reconstruction results of 2D shapes given zero-level set input. Figure 16 shows the reconstruction results of 3D shapes given partial observations.

### D.6.1 NEURAL BRDFS

We show additional qualitative results of BRDF reconstruction in Figure 17. Ours reconstructs correct BRDFs while other baselines fail to reconstruct correct BRDFs resulting in severe color differences in the rendered images. The details are more discernible when the figure is zoomed in.

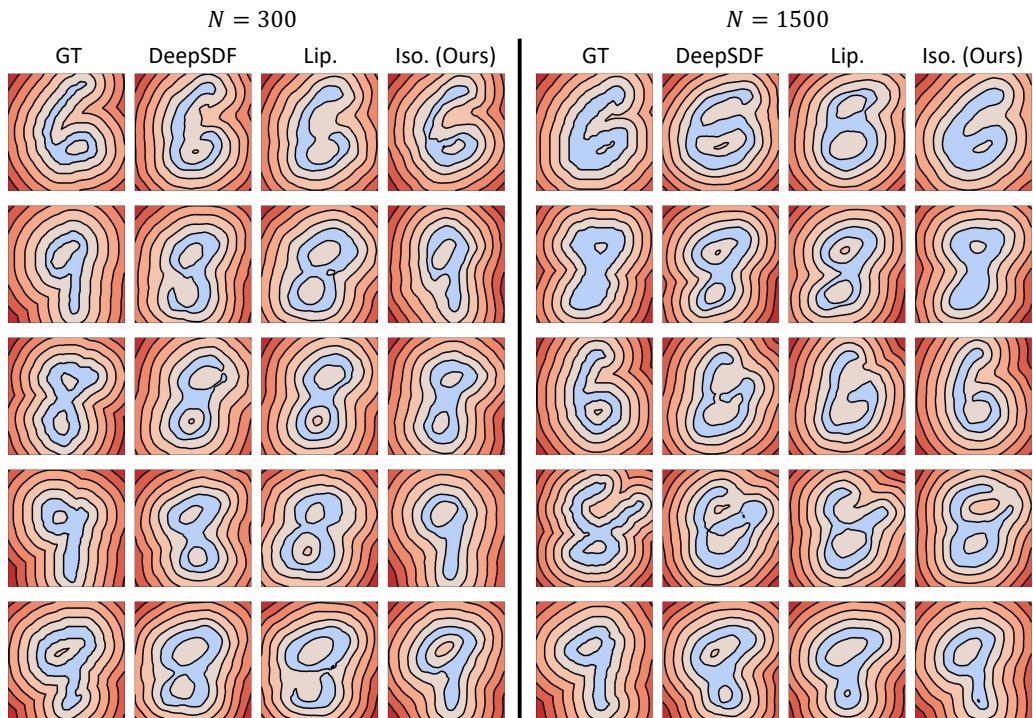

Figure 15: Qualitative results of surface reconstruction given zero-level set on MNIST.

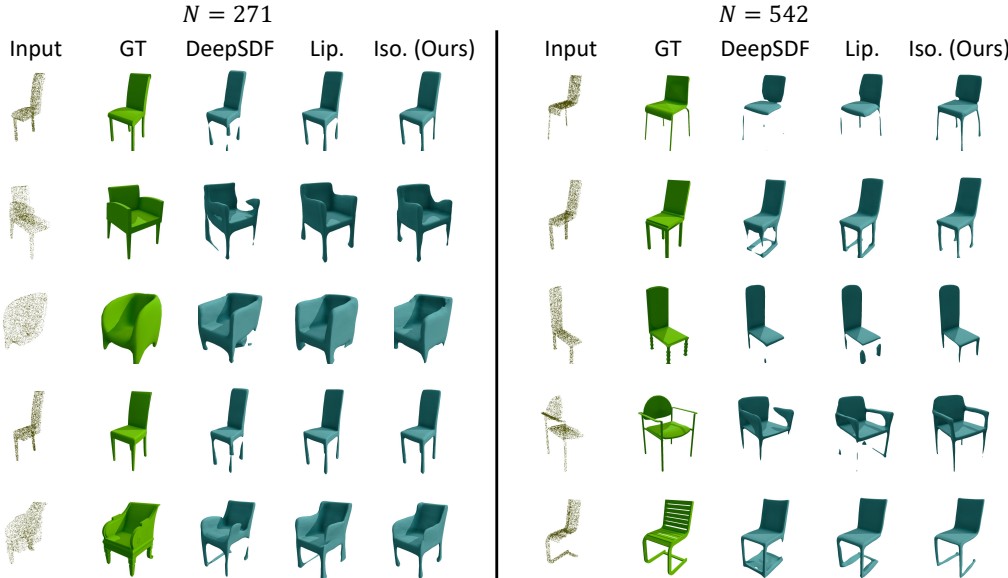

Figure 16: Qualitative results of surface reconstruction on ShapeNet chair dataset.

### D.6.2 NEURAL OPERATORS

We show additional qualitative results of neural operator with reaction-diffusion and Darcy problem datasets in Figure 18 and Figure 19.

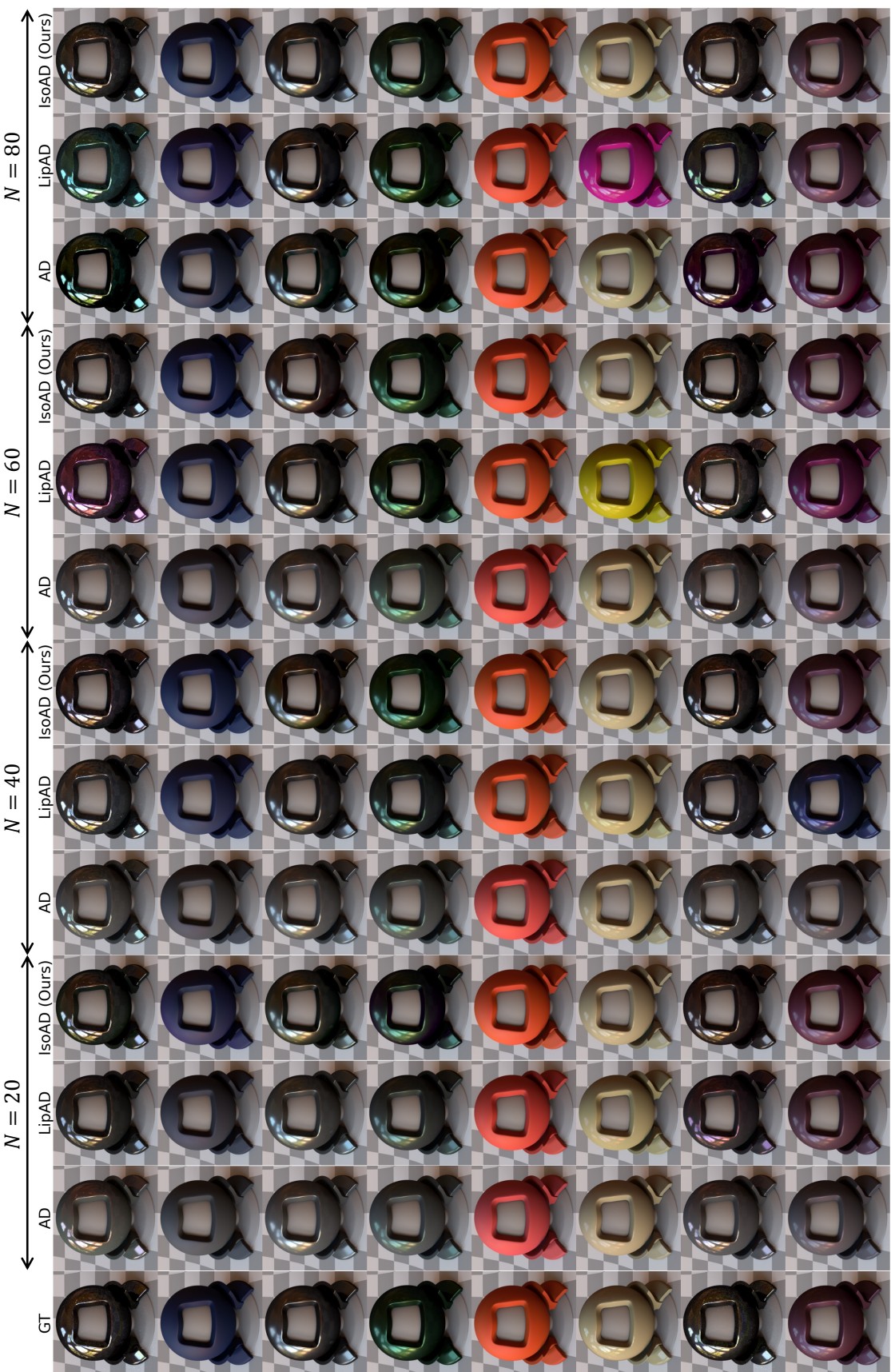

Figure 17: Qualitative results of BRDF reconstruction.

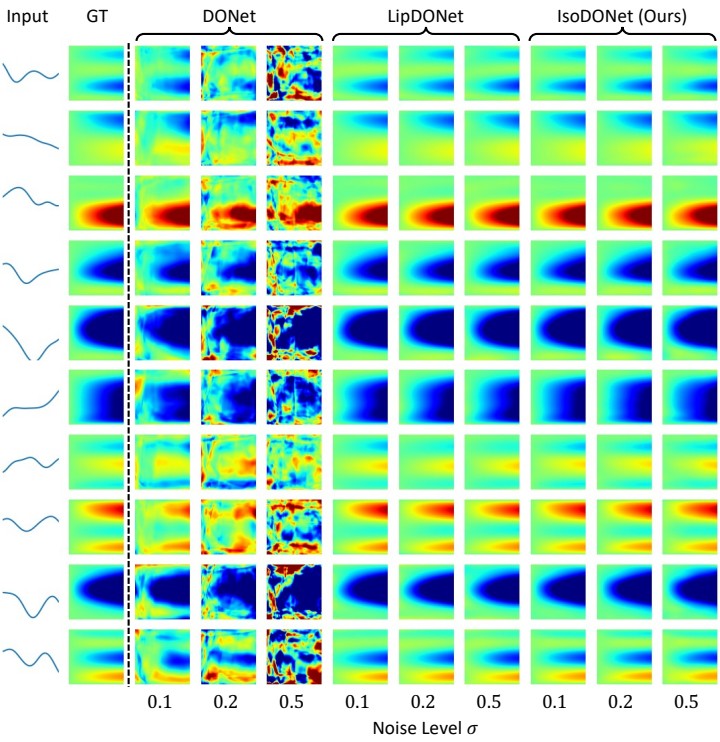

Figure 18: Qualitative results on neural operator with reaction-diffusion dataset.

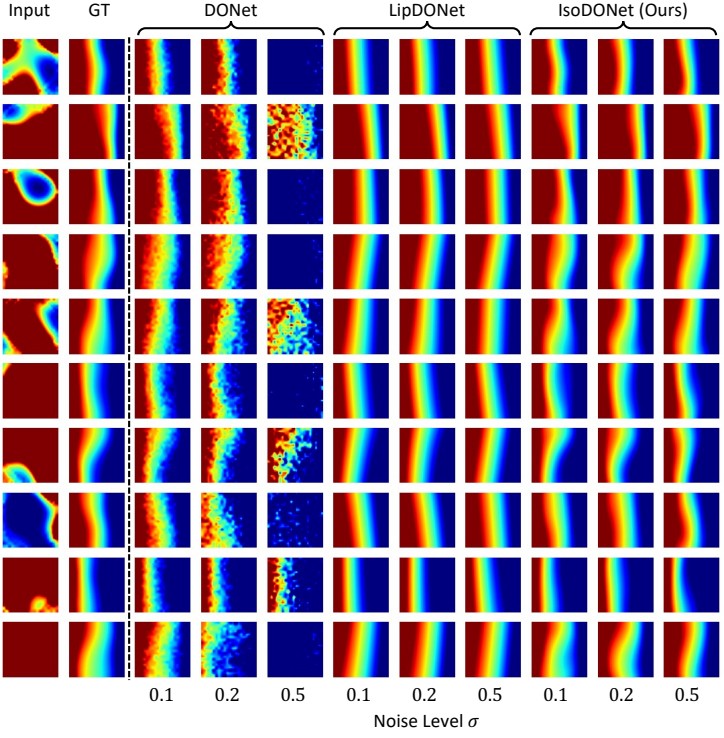

Figure 19: Qualitative results on neural operator with Darcy problem dataset.