# OpenReview forum: "Isometric Regularization for Manifolds of Functional Data"
_ICLR.cc/2025/Conference — ICLR 2025 Poster_

### Official Review · Reviewer_PybU · 2024-10-15

**Soundness:** 4
**Presentation:** 3
**Contribution:** 3
**Rating:** 8
**Confidence:** 4

**Summary:**

The paper proposes a novel isometric regularization technique for Implicit Neural Representations (INRs) aimed at regularizing learned functions, a crucial component of optimizing over a theoretically infinite-dimensional space. This regularization is interpreted through the lens of Riemannian geometry over functional spaces, which helps maintain a well-behaved "latent space" while reducing overfitting, even for small or noisy datasets. The proposed method is validated across multiple scenarios, including synthetic, 2D, and 3D visual data.

**Strengths:**

The proposed method is tested across various types of functional data, including neural SDFs, BRDFs, and neural operators. This cross-domain applicability highlights the generality and versatility of the approach. Experimental results show significant improvements in the quality of latent space, interpolation, and reconstruction tasks. The breath of synthetic to realistic tasks also gives a degree of insight both into the underlying functionality of the method as well as its practical use. The authors also address the computational complexity of infinite-dimensional regularizations and use practical approximations for computational feasibility. Finally, the authors provide an interesting theoretical perspective by interpreting the regularization problem through a Riemannian lens, allowing future researchers to leverage differential geometry principles to derive further regularization algorithms over functional spaces.

In general, this was a concrete, interesting problem with a fresh take that was presented well and demonstrated very well. I enjoyed reading this paper and believe it will make a positive impact on the field.

**Weaknesses:**

1.  While the paper demonstrates robustness and generalizability across several datasets, it does not provide sufficient exploration into the scalability of the approach for very high-dimensional latent spaces or large datasets. An obvious interesting application is in NeRFs, but in my view it is reasonable to delegate this to future work -- there is enough for the paper itself to stand.
2. The paper lacks comprehensive ablation studies on the choices of parameters and effect of the individual components (e.g. the approximations), which may help future researchers in applying and developing from this method.

**Questions:**

1. How does the proposed isometric regularization scale with respect to the dimensionality of the latent space and the number of data points? Have you tested its performance on more complex datasets or tasks beyond those mentioned in the paper?
2. How sensitive is the approach to the choice of hyperparameters, such as the weight of the isometric regularization? Is there a systematic way to select these parameters for different types of data?
3. Can you provide more insights into the computational costs relative to other regularization methods, such as Lipschitz regularization or weight decay, especially for larger datasets?

---

> ### Author Response · Authors · 2024-11-20
>
> **Q1.** How does the proposed isometric regularization scale with respect to the dimensionality of the latent space and the number of data points? Have you tested its performance on more complex datasets or tasks beyond those mentioned in the paper?
>
> **A1.**
> First of all, although it may already be apparent, we would like to emphasize that, from a computational standpoint, our method is scalable to high-dimensional latent spaces and large datasets, thanks to the efficient computation methods we developed for the distortion measure.
> For instance, our Neural SDF experiments employ a 256-dimensional latent space, which is comparably high, as commonly seen in most existing INR studies.
> While we have not yet experimented with more complex datasets beyond those discussed in the paper, exploring such datasets and tasks (e.g., NeRF, as you suggested) would be an exciting direction for future work.
>
>
> **Q2.** How sensitive is the approach to the choice of hyperparameters, such as the weight of the isometric regularization? Is there a systematic way to select these parameters for different types of data?
>
> **A2.** We appreciate the question. Since the scale of the relaxed distortion measure varies across different datasets, selecting the appropriate weight parameter is necessary for each dataset. Empirically, we begin by setting the weight of the regularization term to align with the scale of the original training loss (the loss term between the network output and the training data). We then refine the weight through parameter sweeping to identify the optimal value. More details are provided in Appendix D.2 on regularization weight analysis. Please refer to the details in the revised paper.
>
> **Q3.** Can you provide more insights into the computational costs relative to other regularization methods, such as Lipschitz regularization or weight decay, especially for larger datasets?
>
> **A3.**
> We appreciate and agree with your comment. We have included a comparison of computational time in Appendix D.3.
> The dataset size does not significantly affect the computational complexity of our regularization term. Instead, the complexity of the neural network plays a larger role in determining the computational cost, as the process involves Jacobian-vector and vector-Jacobian products with respect to the network.
> That said, while the computation of our regularization term takes longer, our additional results in Appendix D.4 (which include strategies to further accelerate training) demonstrate that the overall training time is only slightly increased, with our method achieving better performance compared to other baselines, such as Lipschitz regularization.

---

> > ### Comment · Reviewer_PybU · 2024-11-26
> >
> > Thank you for your detailed responses and the included experiments! These definitely help me understand the method a bit more in a practical sense, and I'm sure will also be helpful for readers down the line.
> >
> > I have raised the score -- I do believe this is a strong paper. As the authors have also indicated, there is still some interesting work to do, but this paper does provide a solid foundation for this Riemannian approach for alternative functional regularizations in the age of functional optimizations (e.g. NeRFs).

---

> > > ### Author Response · Authors · 2024-11-27
> > >
> > > Dear Reviewer PybU,
> > >
> > > Thank you for your kind and encouraging comments. We deeply appreciate the time you spent reviewing our work and your thoughtful insights, which helped us improve the clarity and impact of our paper. Your recognition of its contributions and potential is truly motivating.
> > >
> > > Best regards,
> > >
> > > Authors

---

### Official Review · Reviewer_JUq7 · 2024-10-28

**Soundness:** 3
**Presentation:** 2
**Contribution:** 3
**Rating:** 6
**Confidence:** 4

**Summary:**

This paper proposes a way to conditional implicit neural models that may have been conditioned on additional information, such as a class label, or shape vector information. The core idea is to create a mapping from latent-codes to generated output in a way that preserves isometry – that is changes in input and changes in output should be related in a way that preserves distances and angles in the input and output spaces, subject to certain scaling parameters encoded by the function Jacobian. The paper shows experimental results that show the favorable properties of this approach.

**Strengths:**

Geometry-preserving and isometry-preserving loss functions are a novelty, and are a growing area of work in generative models.

The approach to add an isometry-preserving loss to conditional INRs is novel. Approximate algorithms to compute this isometry measure are proposed which enhance the strengths of the paper.

Multiple types of results for proposed INR learning are shown including on 2D, 3D, and neural operator learning.

**Weaknesses:**

Notation is unnecessarily complicated and hinders readability. For instance in the definition of the INR itself, \mathcal{X} is more often than just \mathbb{R}^m with m = 1, 2, or 3. Similarly, V is simply  \mathbb{R}. The lean toward generality isn’t helping with comprehensibility, although everything could be described without loss of generation for f: \mathbb{R}^2 -> \mathbb{R}. I would recommend taking such an approach and moving the generalized framework for an appendix instead. We suffer from the same issue once again when defining the structure of the latent space \mathcal{Z} which always happens to be simply a vector space of the type \mathbb{R}^m.

The premise of section 4, which is the core starts of in a confusing way by stating “Without proper regularization, the latent space can become ill-behaved, overfitting to the data instances, which is exacerbated by the infinite dimensionality of the function space” – here it is unclear what is meant by ‘latent space can become ill-behaved’ – because until this point   \mathcal{Z} is referred to as the latent-space, which is more or less given and not learnt end-to-end. E.g as I understood it,  \mathcal{Z} can refer to class categorical labels, or shape-features that are learnt elsewhere. So I am not at all clear what is meant by ‘latent space can become ill-behaved’.

After reading section 4, we are presented with a way to compute a measure of isometry – as I understand it, this measure really applies to the core INR map, but it is not clear how this measure is actually used in a loss function. No loss function is described from what I can tell. It is unclear thus where latent-codes themselves are being optimized for, or is only the INR mapping portion being optimized. On reading the appendix, it is found that network parameters and the latent codes are both being optimized. I would recommend clarifying this upfront, as we do not see any loss function that suggests latent-code optimization, and this makes it hard to understand how this is being done. This also means that the latent-codes cannot be class categorical labels, as I had initially thought they could be. Overall, this leads to multiple types of confusion which I would suggest really clearly describing.

The paper also uses the term auto-decoder multiple times, which is unclear what that means. Do they mean ‘decoder-only’ architecture, or ‘auto-encoder’. I did additional digging among the references cited to see if I am missing something, but I could not find what an ‘auto-decoder’ means anywhere.

Many conditional INRs also include an element of randomness that creates variation in output, which could be easily included in this model, but was less clear if this was already considered.

Details about training convergence, comparison with other approaches in terms training times, model size would help position the experiments more clearly.

**Questions:**

Please clarify what is the total loss function that is being optimized for, and what parameters are being learnt, and what are being kept fixed?

How is the latent-space itself being learnt?

Discussion about training convergence and model size comparison (without needing new experiments).

Clarify ‘auto-decoder’ terminology.

---

> ### Author Response · Authors · 2024-11-20
>
> **Q1.** Please clarify what is the total loss function that is being optimized for, and what parameters are being learnt, and what are being kept fixed?
>
> **A1.**
> Our total loss function is defined as the sum of the term presented in Equation (9), scaled by a weighting factor, and the original training loss. The original training loss varies depending on the task and typically comprises MSE or L1 loss between the network output and the training data.
> For autoencoders, we optimize the network parameters, whereas for auto-decoders, both the latent codes and network parameters are jointly optimized.
> These details were addressed in our initial draft in Section 5.1, 5.2, 5.3 and Appendix C. To further enhance clarity, we have revised the manuscript to explicitly detail the regularization term within the loss function.
> Please refer to Appendix C for comprehensive training loss details specific to each dataset. If there are any remaining questions or if further clarification is needed, we are more than happy to provide additional information.
>
> **Q2.** How is the latent-space itself being learnt?
>
> **A2.**
> In autoencoders, the latent space is learned through the joint optimization of the encoder and decoder during training. In contrast, for auto-decoders -- originally introduced in [1] -- the latent codes corresponding to the training data points are directly optimized alongside the decoder. It is important to highlight that our method is applicable across both settings.
>
> **Q3.** Discussion about training convergence and model size comparison (without needing new experiments).
>
> **A3.** Our approach builds upon existing INR methods, which have demonstrated stable convergence in prior research. Given that our methodology primarily involves adding a regularization term to these established approaches, we have observed that it does not substantially impact the overall convergence behavior. Thus, we expect our approach to inherit the robust convergence characteristics of existing INR models.
> We use the same network architecture for both models with and without isometric regularization, which leads to the same model size.
>
> **Q4.** Clarify ‘auto-decoder’ terminology.
>
> **A4.** The term ‘auto-decoder’ refers to a ‘decoder-only’ architecture that operates without an encoder component. This terminology was first introduced by Park et al. in their work on DeepSDF [1], and we have adopted it here in alignment with their original usage.
> We also added clarification at the beginning of Section 5.1.
>
> [1] Park et al. "Deepsdf: Learning continuous signed distance functions for shape representation." CVPR 2019.

---

> > ### Comment · Reviewer_JUq7 · 2024-11-24
> >
> > Thank you for the responses; I will consider them.

---

> > > ### Author Response · Authors · 2024-11-27
> > >
> > > Dear Reviewer JUq7,
> > >
> > > Thank you for taking the time to review our work and for considering our responses. We appreciate your efforts and thoughtful evaluation.
> > >
> > > Best regards,
> > >
> > > Authors

---

### Official Review · Reviewer_Awog · 2024-11-01

**Soundness:** 3
**Presentation:** 3
**Contribution:** 3
**Rating:** 6
**Confidence:** 3

**Summary:**

The paper proposes a regularization method, Isometric Regularization, for handling manifolds in implicit neural representations (INRs) designed for infinite-dimensional functional data. It suggests that existing regularization methods often over-smooth data, leading to a loss of fidelity. To address this, the authors introduce a Riemannian manifold-based regularization that minimizes curvature and preserves the geometric consistency between the latent space and data manifold. Experiments across multiple data modalities demonstrate that this approach enhances the structure of the latent space, providing better generalization, especially for noisy and small datasets.

**Strengths:**

1.	The approach is grounded in differential geometry, offering a mathematically rigorous framework for managing functional data representations.
2.	It demonstrated effectiveness on various data types, including neural signed distance functions (SDFs), bidirectional reflectance distribution functions (BRDFs), and neural operators, showcasing its versatility.
3.	The method preserves data fidelity while regularizing, resulting in better interpolation, reconstruction, and generalization across different types and qualities of data.

**Weaknesses:**

1.	This work uses Hutchinson’s stochastic trace estimator to approximate the trace terms, but it is unclear how much additional computational cost this method incurs compared to the baselines. It is better to include a discussion of the computational requirement.
2.	It utilizes offline samples from the "ground truth functional data" to compute the expectation of $J^TJ$. It should provide the justification of accessing the ground truth data.
3.	The regularization process should involve parameter tuning, which might be non-trivial. The paper lacks discussion of how to choose the strength of isometric regularization when applying the method to novel datasets or configurations.

**Questions:**

1.	In Line 416, it says “The effect of regularization is prominent when the training data N is greatly reduced to 20”. However, it is difficult to object the significant improvement of IsoAD with N=20 from Fig. 6? Could you clarify this argument?
2.	How does the strength of regularization choose for different datasets in the paper?

---

> ### Author Response · Authors · 2024-11-20
>
> **Q1.** This work uses Hutchinson’s stochastic trace estimator to approximate the trace terms, but it is unclear how much additional computational cost this method incurs compared to the baselines. It is better to include a discussion of the computational requirement.
>
> **A1.**
> We appreciate and agree with your comment. We have included a comparison of computational time in Appendix D.3 and added experimental results on strategies to further accelerate training in Appendix D.4. While the computation of our regularization term takes longer, our additional results demonstrate that the overall training time is only slightly increased, with our method achieving better performance compared to other baselines, such as Lipschitz regularization.
>
> **Q2.** It utilizes offline samples from the “ground truth functional data" to compute the expectation of $J^TJ$. It should provide the justification of accessing the ground truth data.
>
> **A2.**
> The term "ground-truth functional data" refers to the training data, which we have access to. For instance, in neural SDF examples, we optimize the latent codes $z_i$ and the neural network decoder $ F $ simultaneously. Here, $ F $, evaluated at each latent code $ z_i $ (denoted as $ F_{z\_i} $), is optimized to fit the corresponding training data (i.e., the training SDF function) $F^*\_i  $. By "utilizing offline samples", we mean that instead of sampling from $ p(x; F_{z_i}) $, we use samples from $ F^*\_i $, which $ F_{z_i} $ is being optimized to approximate. This approach avoids the heavy computational load required to sample multiple points from $ F_{z_i} $, which changes at every training iteration. We have clarified this setting by modifying the corresponding paragraph at the end of Section 4. One might question whether this approach is valid. The reason it works is that as training progresses, $ F_{z_i} $ becomes increasingly similar to $ F^*\_i $, making the use of offline samples from $ F^*\_i $ reasonable.
>
> **Q3.**
> The regularization process should involve parameter tuning, which might be non-trivial. The paper lacks discussion of how to choose the strength of isometric regularization when applying the method to novel datasets or configurations.
>
> **A3.**
> Thank you for the valuable comment. As you mentioned, the regularization weight should be carefully selected depending on the datasets and configurations. We have added an analysis of the regularization weight in Appendix D.2 and introduced a strategy to determine the appropriate strength of the isometric regularization.
>
> **Q4.** In Line 416, it says “The effect of regularization is prominent when the training data N is greatly reduced to 20”. However, it is difficult to object the significant improvement of IsoAD with N=20 from Fig. 6? Could you clarify this argument?
>
> **A4.** The performance drop (in terms of PSNR and SSIM) of AD as $N$ decreases is noticeably larger than those with both regularizations, particularly when $N$ is reduced from 40 to 20, as shown in Figure 6. This result demonstrates that the regularization, by smoothing the latent space, improves reconstruction performance, especially when only a limited amount of training data is available. To further clarify, we provide the numbers for Figure 6.
>
> | PSNR         |   N=20 |    N=40 |    N=60 |    N=80 |
> |--------------|------:|------:|------:|------:|
> | AD           | 31.00 | 34.22 | 33.54 | 35.98 |
> | LipAD        | 33.91 | 34.39 | 35.30 | 36.70 |
> | IsoAD (Ours) | 37.35 | 38.48 | 38.65 | 40.62 |
>
> | SSIM         |     N=20 |     N=40 |     N=60 |     N=80 |
> |--------------|-------:|-------:|-------:|-------:|
> | AD           | 0.9286 | 0.9476 | 0.9466 | 0.9544 |
> | LipAD        | 0.9568 | 0.9553 | 0.9625 |  0.972 |
> | IsoAD (Ours) | 0.9731 | 0.9774 | 0.9761 | 0.9882 |
>
> **Q5.** How is the strength of regularization chosen for different datasets in the paper?
>
> **A5.** Since the scale of the relaxed distortion measure varies across different datasets, selecting the appropriate weight parameter is necessary for each dataset. Empirically, we begin by setting the weight of the regularization term to align with the scale of the original training loss (the loss term between the network output and the training data). We then refine the weight through parameter sweeping to identify the optimal value. More details are provided in Appendix D.2 on regularization weight analysis. Please refer to the details in the revised paper.

---

> > ### Comment · Reviewer_Awog · 2024-11-25
> >
> > Thank you for your response. It has resolved my concerns, and I have no additional questions. I will keep the current score.

---

> > > ### Author Response · Authors · 2024-11-27
> > >
> > > Dear Reviewer Awog,
> > >
> > > Thank you for your thoughtful feedback and for letting us know that our responses resolved your concerns. We truly appreciate your time and evaluation of our work.
> > >
> > > Best regards,
> > >
> > > Authors

---

### Official Review · Reviewer_efT4 · 2024-11-01

**Soundness:** 4
**Presentation:** 4
**Contribution:** 2
**Rating:** 6
**Confidence:** 4

**Summary:**

This paper proposes a new regularizer for implicit neural representations (INRs). Motivated by the fact that existing methods such as Lipschitz regularization result in overly smooth neural representations with respect to the spatial coordinates, the authors propose isometric regularization. This regularizer encourages the image of the latent variable model F(x,z) with respect to the latent z variable to form a well-behaved manifold, so that small changes in the latent variable lead to small changes in the resulting INR. As their regularizer requires expensive Jacobian computations at many spatial and latent inputs during training, the authors employ the Hutchinson trace estimator to avoid materializing full Jacobian matrices during training, and pre-sample input points to avoid the need to draw fresh samples during training. They then demonstrate that their method improves on an unregularized baseline and Liu et al’s Lipschitz regularization on surface reconstruction, BRDF learning, and neural operator learning tasks.

**Strengths:**

- The paper is well-written, and the authors do a good job of introducing the mathematical preliminaries needed to describe their method.
- The experiments persuasively make the case that isometric regularization empirically improves on the Lipschitz regularization baseline.
- The numerical scheme the authors propose in Algorithm 1 to avoid costly Jacobian computations and sampling of spatial coordinates at training time is a reasonable strategy for approximating their regularizer.

**Weaknesses:**

I’m not sure if I understand the motivation for isometric regularization relative to simpler alternatives. The authors state that existing works such as Liu et al (2022) apply Lipschitz regularization to the latent variable models F to make them smooth across both the input space X and the latent space Z; this is problematic because for any fixed latent z, the function h(z) is overly smooth in X-space. While I agree that this is a notable deficiency in existing methods, why can one not simply enforce the smoothness of F with respect to the latent coordinates alone?

This question is especially salient in light of the high cost of isometric regularization relative to simpler baselines like Lipschitz regularization. I wonder whether the undeniable improvements arising from using isometric regularization outweigh its costs.

**Questions:**

- How does the computational cost of the proposed method compare to competing methods like LipDeepSDF? I’m particularly interested in understanding how the marginal cost of training the latent variable model with isometric regularization compares to the marginal benefit relative to cheaper baselines.
- Why is it insufficient to enforce the smoothness of F with respect to the latent coordinates alone? Is this especially challenging from a technical standpoint? More challenging than isometric regularization?

---

> ### Author Response · Authors · 2024-11-20
>
> **Q1.** The authors state that existing works such as Liu et al (2022) apply Lipschitz regularization to the latent variable models $F$ to make them smooth across both the input space $\mathcal{X}$ and the latent space $\mathcal{Z}$; this is problematic because for any fixed latent $z$, the function $h(z)$ is overly smooth in $\mathcal{X}$-space. While I agree that this is a notable deficiency in existing methods, why can one not simply enforce the smoothness of $F$ with respect to the latent coordinates alone?
>
> **A1.**
> Lipschitz regularization imposes constraints on the Lipschitz constant of the neural network, defined as $ c = \prod_i \|W_i\|\_{\infty} $, where $ W_i $ is the weight matrix of the $ i $-th layer (assuming the activation function has a maximum slope of 1). This regularization enforces smoothness between the input and output variables across the entire network architecture. However, this approach inherently operates over the combined input spaces, making it difficult to independently control smoothness in $ \mathcal{X} $ and $ \mathcal{Z} $. In fact, as long as the neural network has more than one layer, independently controlling smoothness becomes challenging.
>
> Consider a simple linear map $ f(x, z) = W(x, z) = W_x x + W_z z $. Here, the Lipschitz constant for the $ z $-input can be directly computed as the $ \|W\_z\|\_{\infty} $. However, consider the next simplest case: a two-layer neural network, $ f(x, z) = W_1 a(W_2(x, z)) = W_1 a(W_{2x}x + W_{2z}z) $, where $a(\cdot)$ is an activation. In this case, it is unclear how to control smoothness solely in $ \mathcal{Z} $. If we treat $ \|W_1\|\_{\infty} \|W_{2z}\|_{\infty} $ as the Lipschitz constant for $ z $, minimizing this term would also impact the smoothness in $ \mathcal{X} $ -- because $W_1$ is affected --, making independent control of smoothness in $ \mathcal{Z} $ infeasible. This highlights the inherent difficulty of disentangling smoothness constraints in composite input spaces for multi-layer neural networks.
>
>
> **Q2.** How does the computational cost of the proposed method compare to competing methods like LipDeepSDF? I’m particularly interested in understanding how the marginal cost of training the latent variable model with isometric regularization compares to the marginal benefit relative to cheaper baselines.
>
> **A2.**
> We appreciate your questions and agree that this is an important aspect. First, we have included a comparison of computational time in Appendix D.3.
> Calculating the distortion measure is indeed more computationally expensive compared to Lipschitz regularization, sometimes requiring 2–3 times, or even up to 5 times, more time depending on the problem. However, in practice, we address this by computing isometric regularization term every 5th or 10th iteration instead of at every iteration, which remains effective. Importantly, this approach has shown better performance than Lipschitz regularization applied at every iteration, as demonstrated in the additional experimental results in Appendix D.4.

---

> > ### Comment · Reviewer_efT4 · 2024-11-25
> > **Response to authors' rebuttal**
> >
> > Thank you for your helpful responses to my questions. I understand the difficulty with implementing Lipschitz regularization by directly penalizing the network weights, as you describe in A1. That said, why can't we just use a regularizer of the following form:
> >
> > $R(z) := \sigma^2 \underset{x \sim \mu_{F_z}, \epsilon \sim N(0,I)}{\mathbb{E}} \|J(x,z)\|_F^2$
> >
> > which enforces the smoothness of $F(x,z)$ wrt the latent coordinates $z$ -- but not wrt the $x$-coordinates --  over the support of $\mu_{F_z}$?
> >
> > For small values of $\sigma$, this can be efficiently approximated as follows:
> >
> > $\hat{R}(z) := \underset{x \sim \mu_{F_z}, \epsilon \sim N(0,I)}{\mathbb{E}} [(F(x, z + \sigma \epsilon), F(x,z))^2$],
> >
> > which follows from a first-order Taylor approximation and Hutchinson's trace estimator. This is very cheap to compute, as it does not require any Jacobian computations or JVPs. It seems to me like this strategy, while simple, should enforce the smoothness of $F(x,z)$ wrt $z$ without oversmoothing wrt $x$, and also avoid the heavy costs that you describe in A2. Is there any reason why we wouldn't want to do this? Has this previously been found to be an ineffective strategy?

---

> > > ### Author Response · Authors · 2024-11-26
> > >
> > > Thank you for the excellent question. That is a reasonable proposal; however, minimizing $\|J(x,z)\|_F^2$ proves to be less effective. This is because it can be trivially minimized by increasing the scale of the latent values without affecting the curvature of the actual manifold.
> > >
> > >
> > > For example, consider a re-parameterization map $z \mapsto z' = kz$ for some $k > 1$. Geometrically, this corresponds to a coordinate transformation. Under this transformation, the norm of the Jacobian $\|\frac{\partial F(x,z)}{\partial z}\|_F$ becomes scaled by a factor of $1/k$, while the manifold itself remains identical. This occurs fundamentally because we learn the latent space simultaneously.
> > >
> > >
> > >
> > > This is precisely why, to regularize the manifold curvature (or any geometric properties of the manifold), it is crucial to formulate a geometrically meaningful (i.e., coordinate-invariant) measure. Our relaxed distortion measure, which evaluates how close $F$ is to being a scaled isometry, naturally achieves this goal as it remains invariant under scaling transformations.

---

> > > > ### Comment · Reviewer_efT4 · 2024-12-02
> > > >
> > > > Thank you for answering my question! I appreciate your engagement throughout the discussion period.

---

### Official Review · Reviewer_qgrQ · 2024-11-02

**Soundness:** 3
**Presentation:** 3
**Contribution:** 3
**Rating:** 6
**Confidence:** 3

**Summary:**

This paper proposes an isometric regularization technique for manifolds embedded in an (infinite-dimensional) function space. The goal is to enforce the learned operator $h: Z \to \mathcal{F}$ as an isometric embedding. Here, $Z$ represents a (finite-dimensional) latent space with the standard metric, and $\mathcal{F}$ is the space of functions on $X$, equipped with an appropriately defined metric. A distortion measure that quantifies the lack of isometry (up to rescaling) is introduced, along with an efficient method for its estimation. Numerical experiments validate the effectiveness of the proposed method.

**Strengths:**

1. The paper is well-written, well-organized, and easy to follow.
2. The concept of encouraging isometric embedding from $z$—the latent variable—to $h(z) = F(\cdot, z)$, a functional representation, is interesting.
3. The efficient estimation of a distortion measure that quantifies the lack of isometry is also novel.
4. Extensive experiments, including applications to neural SDFs and DeepONets, effectively demonstrate the robustness of the proposed method

**Weaknesses:**

1. Algorithm 1 and its explanation in Appendix B should be expanded. For instance, further elaboration is needed on estimating the numerator in Equation 9. Additionally, in line 14 of Algorithm 1, should $G$ be given as $G = G_2 / G_1$?
2. Given the significant computation involved in estimating the distortion measure, especially when computing gradients, a comparison of computational time should be included in the paper.
3. While the method promotes an isometric embedding between the latent variable $z$ and its functional embedding $h(z) = F(\cdot, z)$, it remains unclear if an autoencoder mapping the original input $u \in \mathcal{U}$ to its latent code is isometric. For example, in DeepONet, where $u$ represents the input function, if the encoder from $u$ to $z$ diverges substantially from an isometric mapping, enforcing isometry in $z \mapsto h(z)$ may have limited impact.

**Questions:**

Please refer to the previous section

---

> ### Author Response · Authors · 2024-11-20
>
> **Q1.** Algorithm 1 and its explanation in Appendix B should be expanded. For instance, further elaboration is needed on estimating the numerator in Equation 9. Additionally, in line 14 of Algorithm 1, should $\mathcal{G}$ be given as $\mathcal{G}_2/\mathcal{G}_1$?
>
> **A1.**
> Thank you for bringing this to our attention.
> We have revised our manuscript to clarify Algorithm 1 (specifically, line 14 has been updated) and have expanded the explanations in Appendix B.
>
> **Q2.** Given the significant computation involved in estimating the distortion measure, especially when computing gradients, a comparison of computational time should be included in the paper.
>
> **A2.** We appreciate and fully agree with your comments. First, we have added a comparison of computational time in Appendix D.3. As you noted, calculating the distortion measure can be somewhat time-consuming. Therefore, in practice, instead of computing it at every iteration, we compute it every 5th or 10th iteration to apply regularization, which still proves to be effective. Specifically, we have included additional experimental results analyzing this approach in Appendix D.4.
>
> **Q3.** While the method promotes an isometric embedding between the latent variable $z$ and its functional embedding $h(z)=F(:,z)$, it remains unclear if an autoencoder mapping the original input $u\in\mathcal{U}$  to its latent code is isometric. For example, in DeepONet, where represents the input function, if the encoder from $u$ to $z$ diverges substantially from an isometric mapping, enforcing isometry in $z\to h(z)$ may have limited impact.
>
> **A3.** You are correct that the encoder mapping from an input $ u $ to the latent code $ z $ is not explicitly regularized to be isometric. From a practical perspective, the encoder network's inherent smoothness bias likely helps mitigate significant distortions in the latent code's geometry. However, we agree that enforcing the encoder to be closer to an isometric mapping could lead to further performance improvements.
> That said, applying isometric regularization to the encoder poses technical challenges. When the encoder maps a high-dimensional space to a lower-dimensional space, the distortion measure may not be well-defined, as the pull-back metric $ J^TJ $ is not full rank, and such mappings cannot be isometric. A feasible direction might involve identifying a lower-dimensional manifold in the input space and defining a distortion measure between this manifold and the latent space. While this is an interesting future research direction, implementing it would require additional modules and added complexity, which we believe is beyond the scope of this paper.

---

> > ### Comment · Reviewer_qgrQ · 2024-11-25
> >
> > I thank the authors for the detailed response. All my questions have been resolved, and my rating remains.

---

> > > ### Author Response · Authors · 2024-11-27
> > >
> > > Dear Reviewer qgrQ,
> > >
> > > Thank you for your thoughtful feedback and for taking the time to review our work. We are glad that our responses addressed your questions, and we truly appreciate your evaluation throughout the process.
> > >
> > > Best regards,
> > >
> > > Authors

---

### Author Response · Authors · 2024-11-20
**Common Comments**

We would like to express our sincere gratitude for your time and the valuable feedback provided on our manuscript. Your insights and constructive criticism have greatly assisted in enhancing the quality and clarity of our work.

To answer the reviewers' questions and to better clarify our method, we have revised our manuscript and expanded the appendix with the following additional contents:
- D.2 Regularization Weight Analysis
- D.3 Computational Time
- D.4 Regularization Interval Analysis

The changes are highlighted in magenta text within the manuscript.
We provide responses to each reviewer's questions below.

---

### Meta-Review · Area_Chair_RuQ2 · 2024-12-19

**Metareview:**

This paper proposes to imposing isometric constraints in implicit neural representations (INRs) by regularizing the Riemannian metric in an embedded finite-dimensional functional space. It takes a meaningful step to tackle limitations of Lipschitz regularization resulting in over-smooth functional representations. The authors have addressed the computational efficiency and parameter sensitivity issues raised in the reviews.  Therefore, all reviewers are positive in accepting this paper.

If the final version, please carefully consider the reviewers' comments including notational and terminological issues raised by reviewer JUq7, as the paper can still be improved from these aspects. Some of the reviewer's comments (notatinoal issues, and the sentence at the start of section 4, etc.) are not addressed in detail in the rebuttal and should be reflected in the final version.

Additionally, eq.(7), H(z) is already defined after eq.(6). From eq.(8) to eq.(9), please include derivation steps in the appendix. Overall, the quality of the mathematical statements can still be improved.

**Additional Comments On Reviewer Discussion:**

All reviewers are positive of the submission unanimously, and have confirmed they are satisfied with the authors' rebuttal.

A few reviewers are concerned about the computational time, the authors have provided related result in the appendix, showing that the regularizer is more expensive to compute than Lipschitz regularization but still acceptable, and lead to better performance.

Two reviewers asked for parameter sensitivity and ablation study, and the authors have provided a strategy for tuning the regularization strength, and included related studies in the appendix.

Overall, the quality of the mathematical statements can still be improved. I have included two additional remarks for the authors' revision.

---

### Decision · Program_Chairs · 2025-01-22

Accept (Poster)